# Transcription factors ASCL1 and OLIG2 drive glioblastoma initiation and co-regulate tumor cell types and migration

Bianca L. Myers [1], Kathryn J. Brayer[2], Luis E. Paez-Beltran [1], Estrella Villicana [1], Matthew S. Keith[1], Hideaki Suzuki[1], Jessie Newville [1], Rebekka H. Anderson [1], Yunee Lo [1], Conner M. Mertz [1], Rahul K. Kollipara[3], Mark D. Borromeo[4], Q. Richard Lu [5], Robert M. Bachoo[6], Jane E. Johnson [4] & Tou Yia Vue [1,2] ✉

Glioblastomas (GBMs) are highly aggressive, infiltrative, and heterogeneous brain tumors driven by complex genetic alterations. The basic-helix-loop-helix (bHLH) transcription factors ASCL1 and OLIG2 are dynamically co-expressed in GBMs; however, their combinatorial roles in regulating the plasticity and heterogeneity of GBM cells are unclear. Here, we show that induction of somatic mutations in subventricular zone (SVZ) progenitor cells leads to the dysregulation of ASCL1 and OLIG2, which then function redundantly and are required for brain tumor formation in a mouse model of GBM. Subsequently, the binding of ASCL1 and OLIG2 to each other's loci and to downstream target genes then determines the cell types and degree of migration of tumor cells. Single-cell RNA sequencing (scRNA-seq) reveals that a high level of ASCL1 is key in specifying highly migratory neural stem cell (NSC)/astrocyte-like tumor cell types, which are marked by upregulation of ribosomal protein, oxidative phosphorylation, cancer metastasis, and therapeutic resistance genes.

Glioblastoma (GBM) is the most common and malignant form of glioma, with a median survival below 18 months and a 5-year survival rate of less than 10%[1]. A major challenge in the treatment of GBM is that by the time of diagnosis, highly infiltrative tumor cells have already migrated long distances on major white matter tracts and/or the microvasculature to invade surrounding brain regions[2]. Thus, despite the use of fluorescence-guided surgery for maximal resection of tumor tissues followed by concurrent chemotherapy with temozolomide (TMZ) and conformal radiation[3], recurrence is unavoidable since complete elimination of all malignant cells is not possible. The limited efficacy of chemoradiation is also attributed to the heterogeneity and plasticity of GBM cells, whereby glioma stem-like cells (GSCs)

identified via stem cell surface markers have been shown to be highly tumorigenic and resistant to TMZ[4–10]. Currently, the genetic mechanisms that regulate the hierarchy and heterogeneity of GBM tumor cells remain unclear.

Based on bulk RNA sequencing studies, GBMs are classified into three major subtypes—proneural, classical, and mesenchymal—each defined by a unique molecular signature[11–13]. However, single-cell RNA sequencing (scRNA-seq) and multi-location sampling of brain tumors reveal that individual tumors and tumor cells are comprised of multiple GBM subtypes, highlighting a high degree of inter- and intratumoral heterogeneity[14,15]. More recently, single-cell lineage tracing combined with scRNA-seq demonstrates that GBM cells fluctuate

[1]Department of Neurosciences, University of New Mexico Health Sciences Center, Albuquerque, NM, USA. [2]University of New Mexico Comprehensive Cancer Center, Albuquerque, NM, USA. [3]McDermott Center for Human Growth and Development, University of Texas Southwestern Medical Center, Dallas, TX, USA. [4]Department of Neuroscience, University of Texas Southwestern Medical Center, Dallas, TX, USA. [5]Department of Pediatrics, Brain Tumor Center, EHCB, Cincinnati Children's Hospital Medical Center, Cincinnati, OH, USA. [6]Department of Neurology and Neurotherapeutics, University of Texas Southwestern Medical Center, Dallas, TX, USA. ✉e-mail: TVue@salud.unm.edu

between four transient cellular states resembling those of neural progenitor cells (NPCs), oligodendrocyte precursor cells (OPCs), astrocyte cells (AC), and mesenchymal (MES) cells[16]. Although the relative frequencies of these cellular states and GBM subtypes are associated to some extent with specific genetic mutations/alterations or the tumor microenvironment[12,13,16,17], the transcriptomic plasticity of GBM cells is likely an inherent property independent of the types or combinations of driver mutations.

A common feature of GBMs, including lower-grade gliomas, is the presence of multiple neurodevelopmental transcription factors that may be responsible for maintaining tumor cells in an undifferentiated, progenitor-like state[18–20]. Indeed, forced expression of several combinations of core transcription factors can reprogram differentiated glioma cells or transform immortalized astrocytes into tumor-propagating cells when transplanted orthotopically into the brains of immunodeficient mice[18,19]. These studies suggest that the combinatorial functions or differential levels of transcription factors are the direct mechanistic link between oncogenic driver mutations with downstream transcriptional programs that influence or determine the various cellular states and hierarchy of GBM cells. However, which transcription factors are responsible for which cellular states or cell types and how the initiation, proliferation, and migration of GBMs are altered following perturbations of specific sets of transcription factors in vivo remains to be determined.

During development of the central nervous system (CNS), ASCL1 and OLIG2 are two influential transcription factors dynamically co-expressed in various multipotent NPCs, including in highly proliferative and migratory glial progenitor and precursor cells[21–26]. These two bHLH transcription factors are co-expressed in an oscillatory manner with NOTCH signaling and HES proteins, and this oscillation has been proposed to be essential for balancing progenitor cell maintenance with cell fate specification of neuronal, oligodendroglial, and astroglial lineages[27,28]. Similarly, ASCL1 and OLIG2 are highly co-expressed in GBMs, in part because the loci of *OLIG1* and *OLIG2*, along with numerous NSC (*HES5, ID1, ID2, NFIX, SOX2, SOX4, ZEB1*) and glial lineage transcription factors (*ID3, NFIA, NFIB, NKX2-2, SOX8, SOX9, SOX10*), are major targets of ASCL1 binding[20,29]. Developmentally, GBM cells follow a roadmap similar to that of the fetal brain, where ASCL1 marks the presence of highly proliferative, TMZ-resistant glial progenitor (GPC) cancer cells at the apex of astroglial, oligodendroglial, neuronal, and mesenchymal cancer cells[30]. This similarity implies that, as seen during neurogenesis and gliogenesis, the dynamic levels of ASCL1 and OLIG2 may directly underlie the process of gliomagenesis and the lineage composition of GBMs.

In this study, we show that ASCL1 and OLIG2 can physically and genetically interact to bind to similar sites in the genome of two separate patient-derived-orthotopic GBM xenografts (PDOX-GBMs) grown in the brains of NOD-SCID mice. By combining sophisticated transgenic gain- and loss-of-function strategies with an innovative CRISPR-Cas 9 GBM mouse model in which fluorescently labeled brain tumors are induced from radial glia in the dorsal SVZ, we demonstrate that ASCL1 and OLIG2 are prominent drivers of brain tumor initiation, proliferation, migration, and cell type specification. Notably, ASCL1 and OLIG2 sit at the apex of the GBM cellular hierarchy, where their dynamic co-expression and functional interactions are directly responsible for determining NSC/astrocyte-like and OPC-like tumor cells, respectively.

## Results

### ASCL1 and OLIG2 share extensive overlap in binding in the genome of orthotopic GBM xenografts

To understand the combinatorial function of ASCL1 and OLIG2 in GBMs, we performed ChIP-seq for OLIG2 for direct comparison with previously published ASCL1 ChIP-seq in two PDOX-GBM lines[20]. We found that OLIG2 binds to 105,741 statistically significant sites (green

circle, Fig. 1a) and overlapped with over 90% of the 13,457 ASCL1 binding peaks (green and blue circle overlap, Fig. 1a, b; Supplementary Data 1). De novo motif analyses showed a 65% enrichment of E-boxes with a strong "GC" core (CAGCTG) within the ASCL1 and OLIG2 shared binding peaks, whereas E-boxes with variable "GC" or "TA" core were observed within 50% of the OLIG2 only binding peaks. This difference in E-box enrichment and consensus sequences was also accompanied by enrichment of different DNA co-factor motifs for ASCL1 and OLIG2 shared binding peaks (SOX8, ZFX) versus OLIG2 only binding peaks (PR, BMYB) (Fig. 1c). These differences may explain the differential binding specificity of ASCL1 and OLIG2 in the genome of GBMs.

To identify putative transcriptional targets of ASCL1 and OLIG2, we next used Genomic Regions Enrichment of Annotations Tool (GREAT)[31] to associate the binding peaks of ASCL1 and OLIG2 with the nearest genes[20]. ASCL1 binding peaks were associated with 8,883 target genes, whereas OLIG2 binding peaks were associated with 16,022 target genes (Supplementary Data 2). Roughly 98% of the ASCL1 target genes were also targets of OLIG2 (Fig. 1d). Next, we applied Spearman rank-ordered correlation (>0.4) to identify the top genes positively correlated with both *ASCL1* and *OLIG2* in RNA-seq of 164 TCGA GBM samples (Supplementary Data 3, 4)[11,20]. This produced a total of 1580 genes, 841 of which were direct targets and thus likely to be regulated by ASCL1 and OLIG2 binding (purple circle, Fig. 1d). Gene Ontology (GO) analysis of these 841 shared targets revealed that about 40% (348) were associated with function within the nucleus, many of which play direct roles in transcription regulation and DNA binding. Additionally, about 7% (63) were associated with the cell cycle and are responsible for promoting cell division and mitosis (Fig. 1e; Supplementary Data 5). These functions are consistent with the proposed roles for ASCL1 and OLIG2 as core regulators of tumor-propagating cells[11,19,20,30]. Interestingly, we identified 539 genes that were negatively correlated with *ASCL1* and *OLIG2*, 211 of which are direct targets of ASCL1 and OLIG2 shared binding (red circle, Fig. 1d). GO analysis of these 211 negatively correlated shared targets showed significant enrichment of genes associated with immune responses and inflammation (Fig. 1f; Supplementary Data 6). These results suggest that while ASCL1 and OLIG2 promote the tumorigenicity of GBM cells, they may also suppress activation of immune cells in the tumor microenvironment to further support tumor growth.

We next investigated if ASCL1 and OLIG2 interact at both the genetic and protein levels, given their extensive shared binding as bHLH transcription factors. ChIP-seq analyses revealed the presence of strong OLIG2 binding peaks at the *Ascl1* locus, significant ASCL1 and OLIG2 shared binding peaks at *OLIG1/2* loci, and loci of NOTCH signaling (*DLL1, DLL3, NOTCH1, HES1, HES5, HES6*) and NSC target genes (*INSM1, ID1, ID2*) (TS2). ASCL1 and OLIG2 binding peaks were also found at loci of *TCF3, TCF4, and TCF12* genes (Fig. 1g), which encode for bHLH E-protein DNA co-binding partners of ASCL1 and OLIG2[32]. Co-immunoprecipitation (co-IP) assay demonstrated that the shared binding between ASCL1 and OLIG2 may be due to a direct protein-protein interaction since OLIG2 was successfully pulled down with an antibody specific for ASCL1 from PDOX-GBM cells (Fig. 1h, i). Collectively, our ChIP-seq and co-IP analyses suggest that ASCL1 and OLIG2 co-regulate an NSC transcriptional network pertinent to promoting gliomagenesis.

### ASCL1 and OLIG2 play redundant roles in tumor initiation but have opposing impacts on tumor cell migration in the brain of a GBM mouse model

The requirement of ASCL1 or OLIG2 was previously tested in two separate GBM mouse models[20,33]. Notably, loss of either ASCL1 or OLIG2 had only modest effects on tumor progression and survival, likely due to redundant functions of these two transcription factors given their extensive shared binding in GBMs (Fig. 1). Here, we hypothesized that the loss of both ASCL1 and OLIG2 should prevent or significantly

**Fig. 1 | ASCL1 and OLIG2 physically interact and overlap in binding in genome of PDOX-GBMs. a** Venn diagram of ChIP-seq binding peaks for ASCL1 or OLIG2 from two different PDOX-GBMs (R548, R738). **b** Heatmap of ASCL1 and OLIG2 shared binding sites. **c** DNA sequence motifs that are enriched within ASCL1&OLIG2 or OLIG2 alone binding peaks. Percentage represents the frequency of indicated DNA motif found within 150-bp peak summits compared to (percentage) frequency of that motif in background genomic sequence along with *p*-value. Motif enrichment is calculated using cumulative binomial distributions. **d** Overlap of genes associated with ASCL1 or OLIG2 binding peaks intersecting with genes positively or negatively correlated with *ASCL1* and *OLIG2* expression in RNA-seq of 164 GBM samples from the TCGA. **e** Gene ontology analysis of 841 positively correlated genes showing enrichment of molecular functions in the nucleus and cell cycle. **f** Gene ontology analysis of 211 negatively correlated genes showing enrichment of molecular functions in immune response and inflammation. Significance of enrichment was determined using Fisher's Exact test. **g** ASCL1 and OLIG2 binding peaks at the other's loci and known NOTCH (*DLL3, NOTCH1, HES5*), NSC (*INSM1*), and bHLH E-protein (*TCF3, TCF4*) targets. **h** Dynamic co-localization of ASCL1 and OLIG2 in PDOX-GBM. Percentage co-localization of ASCL1 and OLIG2 in PDOX-GBM was previously reported[20]. Scale bar: 25 μm. **i** Co-IP assay of PDOX-GBM using anti-ASCL1 antibody. Immunoblot (IB) showing presence of OLIG2 (~40 kDa, red arrowhead) in IP lane. Note that the 55 kDa band in ASCL1 IB panel is IgG. This result was observed in two independent experiments.

reduce brain tumor formation and/or progression, resulting in an increase in survival. To test this, we used a CRISPR-Cas 9 transgenic GBM mouse model in which gliomas are induced and fluorescently labeled in the brains of immunocompetent mice. Specifically, plasmids expressing Cre-recombinase and Cas 9 + guide RNAs targeting *Tp53,* *Nf1,* and *Pten* (TNP)[34] were injected and then electroporated into progenitor cells lining the dorsal SVZ of the right lateral ventricle of Cre-dependent tdTomato (tdTOM) reporter mice (*R26R^{T/T}*)[35] at birth (P0) (Fig. 2a; Supplementary Fig. 1a–d). The majority of electroporated tdTOM+ cells do not express ASCL1 or OLIG2 and are likely radial glia,

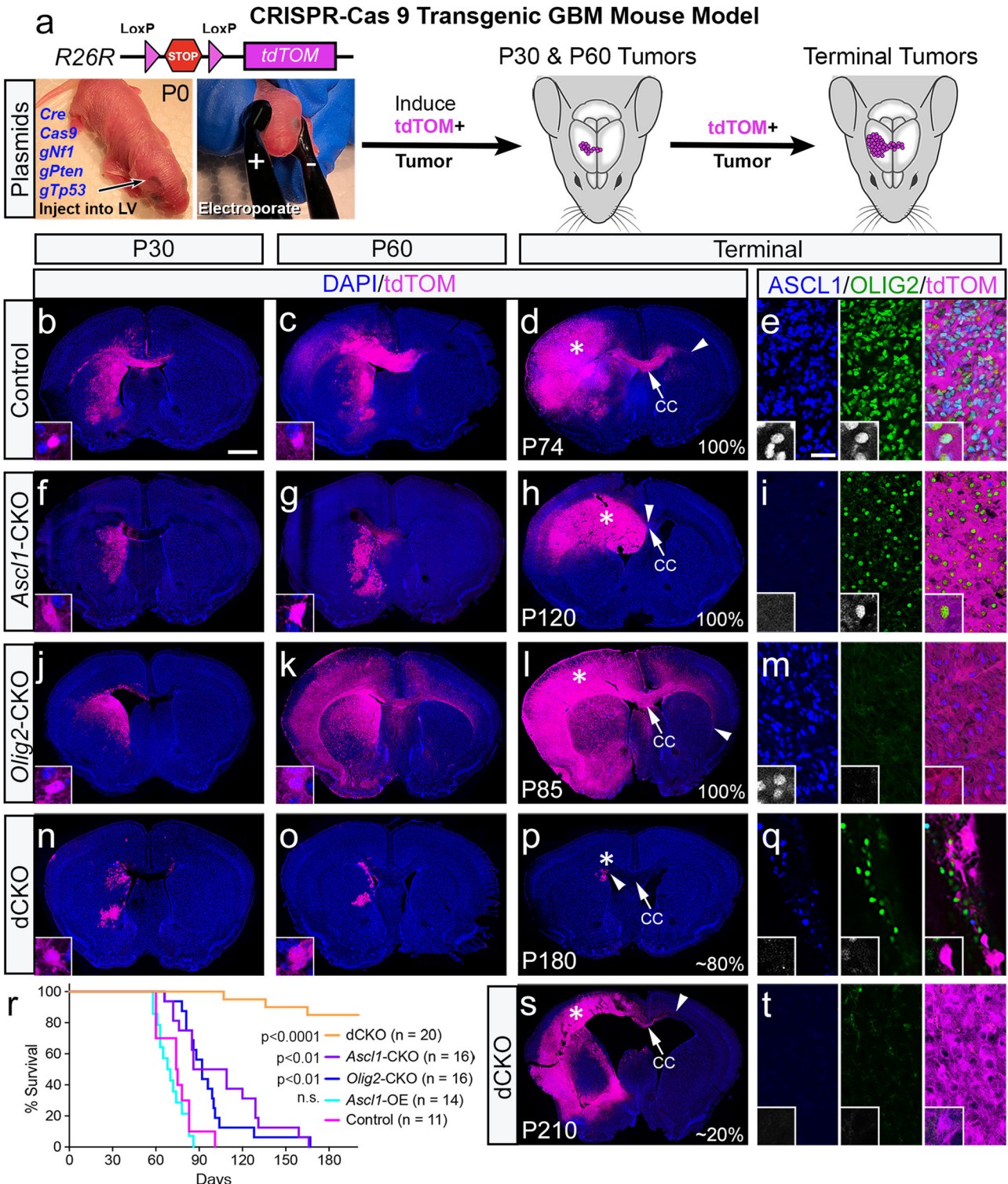

**Fig. 2 | ASCL1 and OLIG2 are required for tumor formation but inversely regulate different aspects of tumor migration in GBM mouse model. a** Schematic showing induction of tdTOM+ brain tumors via electroporation of indicated Cre + CRISPR plasmids into neural progenitor cells in the right lateral ventricle of *R26R*[T/T] reporter mice at birth for longitudinal analyses. (**b–q, s, t**) Representative images of tdTOM+ tumors at P30, P60, or terminal stage in control (**b–e**), *Ascl1*-CKO (**f–i**), *Olig2*-CKO (**j–m**), and double CKO (**n–q, s, t**) mice (number of tumors imaged: *n* = 4/genotype for P30 & P60 and *n* = 6/genotype for terminal tumors). Arrows indicate midline and arrowheads mark the distance of migration of tdTOM+ tumor cells on the contralateral corpus callosum (CC). Asterisks demonstrate region imaged for (**e, i, m, q,** and **t**). ASCL1 and OLIG2 are highly co-expressed in control tdTOM+ tumor cells, but absent in the single or double CKO tdTOM+ tumors. Scale bars: 1 mm for whole brain sections; 25 μm for (**e, i, m, q, t**), and 12.5 μm for all insets. **r** Kaplan-Meier survival curves of each group of tumor mice showing statistical significance (Mantel-Cox test) between control versus experimental groups.

as evidenced by long radial processes in the cortex of postnatal day 6 (P6) mice (Supplementary Fig. 1e, f). With TNP-deletion, 100% of electroporated mice consistently developed tdTOM+ tumors surrounding the right lateral ventricle in the striatum, corpus callosum, and cortex, which co-expressed high levels of ASCL1 and OLIG2 and died between 2-3 months of age (Fig. 2b–e, r). Interestingly, deletion of one or any two of the TNP genes fails to induce full-blown lethal tumors even by six months (P180) of age (Supplementary Fig. 1g), highlighting the synergistic effects of these somatic mutations on GBM development. The tdTOM+ tumors were invasive, capable of migrating across the corpus callosum to the contralateral hemisphere, and exhibited histopathological characteristics of hypercellularity, nuclear atypia, mitotic cells, and pseudopalisading necrosis similar to human GBMs (Supplementary Fig. 1h). We then bred Cre-dependent floxed allele of Ascl1[36] (Ascl1[F/F];R26R[T/T]), Olig2[37] (Olig2[F/F];R26R[T/T]), or both of these alleles (Ascl1[F/F];Olig2[F/F];R26R[T/T]) into tdTOM reporter mice to perform conditional knock-out (CKO) studies to directly test the requirement of these two transcription factors specifically within brain tumors.

As in a previous study[20], we found that Ascl1-CKO did not prevent tumor formation but did increase median survival time (98 days versus 75 days for control mice) (Fig. 2r). This increase was exhibited by smaller tdTOM+ brain tumors at P30 and P60 (Fig. 2f, g). Interestingly, there was a noticeable reduction in tdTOM+ tumor migration along the corpus callosum to the contralateral hemisphere, even at terminal stages, although OLIG2 was still present in these tumors (Fig. 2h, i). Overall, the migration distance of Ascl1-CKO tdTOM+ tumors on the contralateral corpus callosum was decreased by 50% compared to control tumors (Fig. 3i). This finding indicates that ASCL1 is a positive regulator of tumor cell migration in the brain.

Similar to Ascl1-CKO, Olig2-CKO resulted in 100% tumor formation with a median survival of 92 days (Fig. 2r). ASCL1 was also expressed in these tumors (Fig. 2m). However, unlike Ascl1-CKO tumors, Olig2-CKO tumors were highly diffuse. Indeed, from P60 to terminal stages, Olig2-CKO tumors migrated extensively to occupy both the ipsilateral as well as contralateral corpus callosum and cortex (Fig. 2j–l). This dramatic migration was accompanied by a significant decrease in tumor cell density on the ipsilateral hemisphere and a 2- to 4-fold increase in migration distance on the contralateral corpus callosum compared to control and Ascl1-CKO tumors (Fig. 3h, i). Thus, unlike ASCL1, this suggests that OLIG2 may function as a suppressor of tumor cell migration.

As predicted, double CKO (dCKO) of both Ascl1 and Olig2 prevented tumor formation in approximately 80% of TNP-deleted mice (N = 16/20, 4 litters) (Fig. 2p, q), while 20% developed slow-growing tumors that were not lethal until 4-6 months of age (Fig. 2r–t). This finding is consistent with the redundant function of ASCL1 and OLIG2 in brain tumor initiation and progression. Indeed, compared to control or single CKO tumor mice, there were far fewer dCKO tdTOM+ cells surrounding the right ventricle in the striatum and corpus callosum at P30 or P60, and the number of cells decreased with age (Fig. 2n, o). Furthermore, these dCKO tdTOM+ cells failed to migrate onto the corpus callosum or the overlying cortex in the non-tumor mice even by six months of age (Fig. 2n–p). A few of the dCKO lethal tumors that developed did show extensive ipsilateral migration into the cortex and some contralateral migration on the corpus callosum (Fig. 2s), but this migration was not as drastic compared to control or Olig2-CKO tumors (Fig. 2d, l).

We also analyzed whether sex had an effect on tumor development and found that there was no statistical difference in survival between males and females for control or the single and double CKO tumor mice (Supplementary Fig. 1i, j). Taken together, our findings imply that ASCL1 and OLIG2 function redundantly downstream of TNP-deletion to transform affected radial glia into proliferating tumor cells, but these transcription factors seem to regulate opposing aspects of tumor cell migration in the brain.

## High levels of ASCL1 promote a highly migratory and diffuse glioma phenotype

The fact that ASCL1 and OLIG2 can physically and genetically interact (Fig. 1g–j) suggests that OLIG2 may directly repress ASCL1's ability to promote tumor migration through these interactions. We hypothesized that if ASCL1 is required to promote tumor migration, then increasing the levels of ASCL1 should overcome the repression by OLIG2, resulting in a highly migratory and diffuse tumor phenotype similar to Olig2-CKO tumors. To test this, we induced tumors in transgenic mice carrying dual alleles of a Cre-dependent tetracycline transactivator (R26R[tTA])[38] and a TetO-promoter driving expression of Ascl1-ires-GFP[39] (Fig. 3a), which resulted in efficient induction of Ascl1-overexpression (OE) GFP+ tumors. Analysis of cellular immunofluorescent intensity of GFP+ tumor cells confirmed that the level of ASCL1 was elevated by about 2-fold compared to control or Olig2-CKO tumor cells (Fig. 3f). The median survival of Ascl1-OE tumor mice (72 days), though not different from control, was significantly shorter (p < 0.0001) compared to Ascl1-CKO and Olig2-CKO tumor mice (Fig. 2r). Similar to the PDOX-GBMs, co-IP assay showed that OLIG2 was successfully pulled down from Ascl1-OE tumors with an antibody specific for ASCL1, but this interaction was lost in Ascl1-CKO tumors (Fig. 3j).

As expected, GFP+ tumor cells migrated extensively on the ipsilateral corpus callosum and cortex at P30 (Fig. 3b). By P60 to terminal stages, GFP+ tumor cells infiltrated the striatum and cortex of both ipsilateral and contralateral hemispheres (Fig. 3c, d), similar to Olig2-CKO tumors. Indeed, the migration distance of Ascl1-OE GFP+ tumor cells on the contralateral corpus callosum was similar to that of Olig2-CKO tumors (Fig. 3i). However, despite this similarity, the immunofluorescent levels of ASCL1 in Olig2-CKO tumor cells were slightly lower than control and significantly lower compared to Ascl1-OE tumor cells (Fig. 3f). Conversely, the immunofluorescent levels of OLIG2 were lowered in Ascl1-OE tumor cells compared to Ascl1-CKO tumor cells (Fig. 3g). These findings imply that the highly migratory property of GBM cells is due to the imbalance of a higher level of ASCL1 to OLIG2.

## ASCL1 and OLIG2 are highly co-expressed and function redundantly to promote tumor cell proliferation

Due to the ability of ASCL1 and OLIG2 to directly bind to the other's loci and loci of cell cycle genes (Fig. 1e, g), we next investigated how alteration of the levels of ASCL1 and/or OLIG2 affect each other's expression and tumor cell proliferation at both early and terminal stages of the various tumor types. Within control tumors, we found that 74% of tdTOM+ tumor cells were OLIG2+ at P30 but were decreased to 61% at terminal stages (Supplementary Fig. 2a, b, k). In contrast, only 36% of control tdTOM+ tumor cells were ASCL1+ at P30, which then increased to 54% at terminal stages (Supplementary Fig. 2e, f, l). Co-localization analysis of ASCL1 and OLIG2 double+ cells in terminal tumors showed that while only 64% of OLIG2+ tumor cells are ASCL1+, 86% of ASCL1+ tumor cells were OLIG2+ (Supplementary Fig. 2m). Interestingly, Ascl1-CKO or Olig2-CKO significantly reduced the percentage of tumor cells expressing the other transcription factor at P30 compared to control (Supplementary Fig. 2c, g, k, l), but this reduction was not observed in terminal tumors (Supplementary Fig. 2d, h, k, l). In contrast, Ascl1-OE significantly increased the percentage of OLIG2+ tumor cells in both P30 (89%) and terminal (87%) tumors compared to controls (Supplementary Fig. 2i–k; dark green bars). This high rate of co-localization of OLIG2 within ASCL1+ cells in control and Ascl1-OE tumors strongly indicates that Olig2 is a direct transcriptional target of ASCL1.

All tumor mice were injected with EdU, a thymidine analog that is incorporated into replicating cells during S-phase, for 2 h prior to tumor harvest; therefore, we analyzed regions of brain tumors with the highest density of EdU+ cells for all tumor types at P30 and terminal stages. Within control tumors, 5% of tdTOM+ tumor cells were EdU+ at

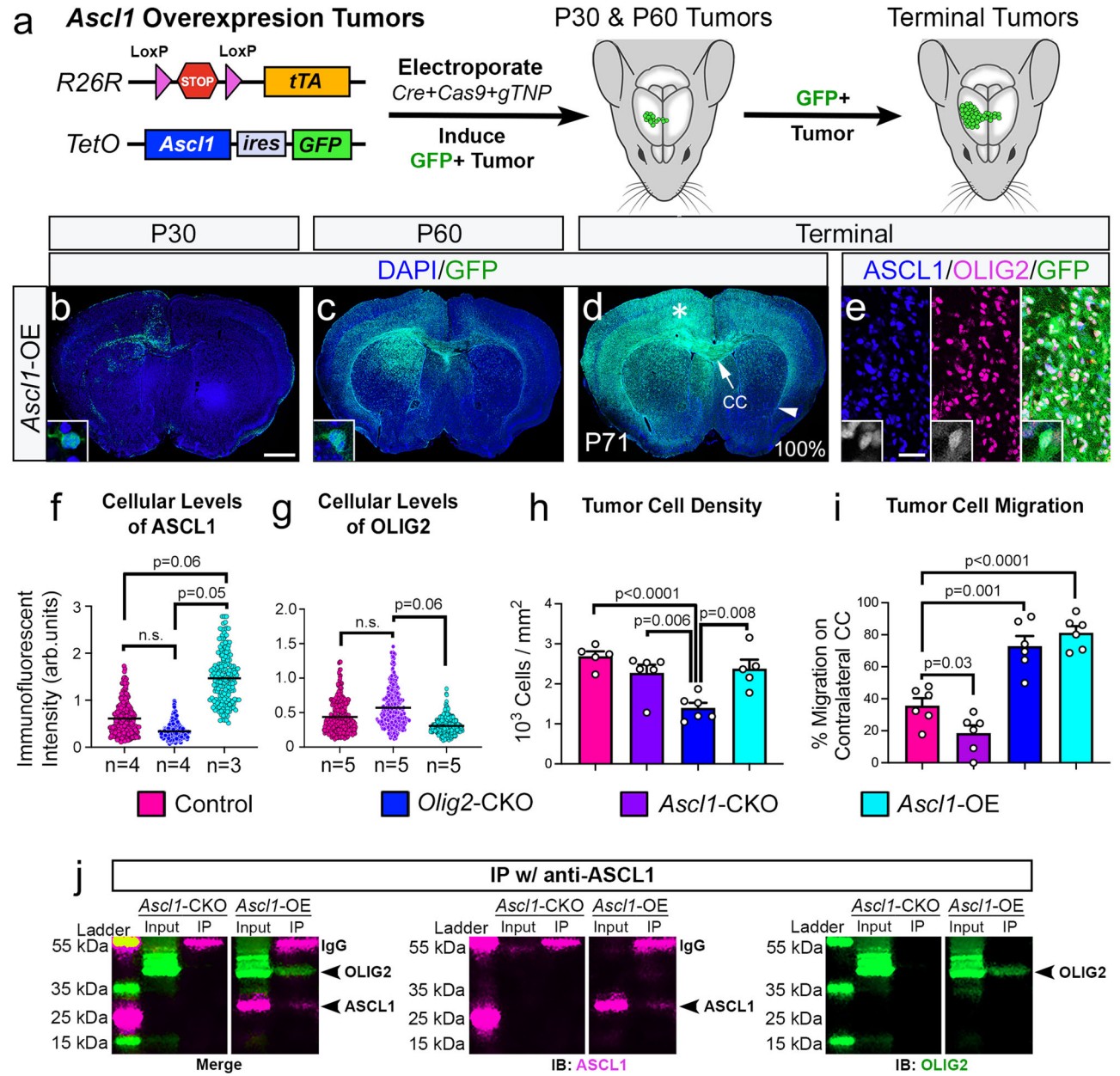

**Fig. 3 | ASCL1 overexpression (OE) promotes tumor cell migration.**
**a** Schematic induction of *Ascl1*-OE GFP+ tumor model. Cre-mediated recombination following electroporation results in sustained expression of tetracycline transactivator (tTA), which binds to *TetO*-promoter to drive expression of *Ascl1* and *GFP*. **b**–**e** Representative images of P30, P60, and terminal-stage tumors highlighting extensive migration and co-expression of ASCL1 and OLIG2 in GFP+ tumor cells. Asterisk indicates region imaged for (**e**). Arrow indicates midline and arrowhead marks the distance of migration of GFP+ tumor cells on the contralateral CC. Scale bars: 1 mm for (**b**–**d**); 25 μm for (**e**); and 12.5 μm for all insets. **f, g** Scatter plot of immunofluorescent intensity of ASCL1 or OLIG2 within individual DAPI+ nuclei of tumor cells of genotypes indicated (*n* represents the number of mice). Statistical significance is determined by comparing the mean immunofluorescent intensity of ~60 tumor cells/mouse/genotype using unpaired *t*-tests with Welch's correction.

**h, i** Quantification of the density of DAPI+ tumor cells in tumor bulk, and distance of migration of reporter+ tumor cells on contralateral CC normalized to the total length of the contralateral CC. Data shown as mean ± SEM. Statistical significance is determined by comparing the means of tumor types using unpaired *t*-tests with Welch's correction. (**h:** control *n* = 5 mice, *Ascl1*-CKO n = 5 mice, *Olig2*-CKO *n* = 6 mice, *Ascl1*-OE *n* = 6 mice; **i:** *n* = 6 mice/genotype). **j** Co-IP assay of *Ascl1*-OE and *Ascl1*-CKO tumors using anti-ASCL1 antibody. Immunoblot (IB) showing presence of OLIG2 (~40 kDa) in IP lane of *Ascl1*-OE tumor but not in *Ascl1*-CKO tumor. Note that ASCL1 (~32 kDa) is detected in both Input and IP of *Ascl1*-OE tumor but not in *Ascl1*-CKO negative control tumor, demonstrating specificity of anti-ASCL1 antibody. This result was observed in two independent experiments. Source data are provided as a Source Data File.

P30, and this was increased to 9% at terminal stages (Supplementary Fig. 3a, b, k, l). Interestingly, neither *Ascl1*-CKO nor *Olig2*-CKO had a significant impact on tumor cell proliferation at P30 or terminal stages (Supplementary Fig. 3c–f, k, l). In contrast, dCKO showed no incorporation of EdU within tdTOM+ cells at P30 or even in the few terminal-stage dCKO tumors (Supplementary Fig. 3g, h, k, l), indicating that tumor cell proliferation was significantly compromised in the absence

of both ASCL1 and OLIG2. Conversely, *Ascl1*-OE GFP+ tumor cells had a significantly higher percentage of EdU+ cells at P30 (15%) and terminal stage (12%) compared to control and/or the single CKO tumors (Supplementary Fig. 3i–l).

Taken together, these findings demonstrate that ASCL1 and OLIG2 are differentially dysregulated by the loss of *Tp53*, *Nf1*, and *Pten* at early stages (P30), but these transcription factors can reciprocally and

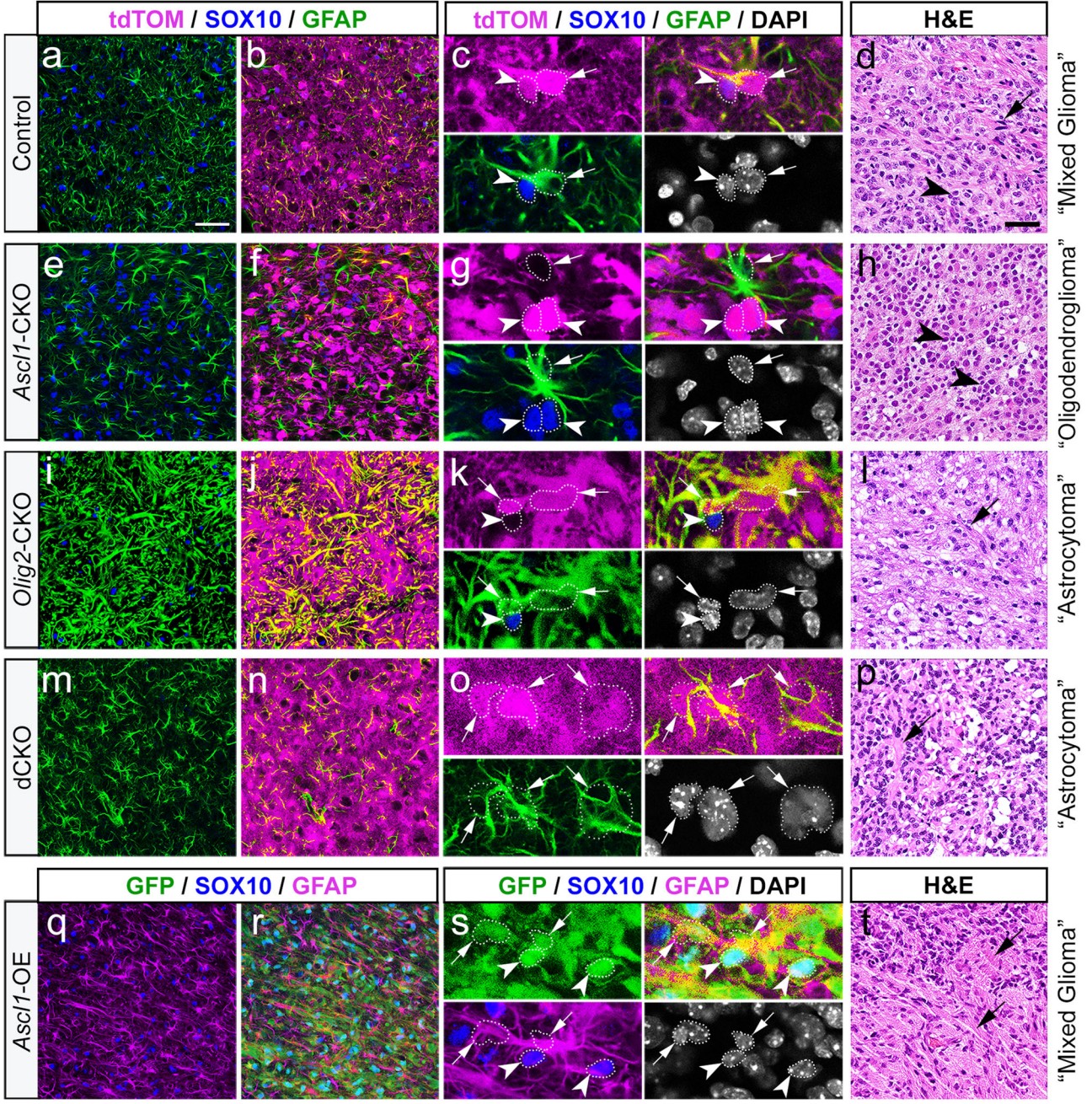

**Fig. 4 | ASCL1 and OLIG2 inversely regulate glioma tumor cell types.** Representative immunofluorescent images and H&E staining of terminal control (**a**–**d**), *Ascl1*-CKO (**e**–**h**), *Olig2*-CKO (**i**–**l**), dCKO (**m**–**p**), and *Ascl1*-OE (**q**–**t**) tumors. High magnification showing differences in co-localization, or lack thereof, of GFAP (white arrow) or SOX10 (white arrowhead) in reporter+ tumor cells. Dotted line delineates DAPI+ cell nuclei that overlap with tumor reporter and/or cell type markers. Similar staining was seen in *n* = 3/genotype. Black arrows and arrowheads in H&E panels indicate the presence of GFAP+ pink fibers and "fried-egg" cell shapes characteristic of astrocytoma and oligodendroglioma, respectively. Scale bars: 12.5 μm for (**c**, **g**, **k**, **o**, **s**) and 50 μm for all other panels.

positively regulate or influence the other's expression within growing brain tumors. Furthermore, ASCL1 and OLIG2 function redundantly to promote tumor formation and proliferation.

## ASCL1 and OLIG2 play inverse roles in regulating glioma tumor cell types

Based on their developmental roles in glial cell fate specification[21–26], we hypothesized that altering the levels of ASCL1 and/or OLIG2 should influence the cell or tumor types of the GBM mouse model. Analysis of whole brain sections showed that control tumors (tdTOM+) typically contained areas of high GFAP (arrows) and dense regions of SOX10+ cells (arrowheads) (Supplementary Fig. 4a–d).

Higher magnification images confirmed that tdTOM mostly colocalized with either GFAP or SOX10 (Fig. 4a–c), indicating that they are "mixed gliomas" comprising of both astroglial or oligodendroglial lineages, respectively. Interestingly, *Ascl1*-CKO resulted in the formation of highly dense SOX10+ "oligodendrogliomas" (arrowheads, Supplementary Fig. 4g, u; Fig. 4e–g). Although GFAP is found within the bulk of *Ascl1*-CKO tumors (Supplementary Fig. 4f), the majority of GFAP+ cells are not tdTOM+ and thus likely to be reactive astrocytes (arrow, Fig. 4e–g). Conversely, *Olig2*-CKO tumors are highly diffuse "astrocytomas" dominated by high levels of GFAP, which overlaps extensively with tdTOM (arrows, Supplementary Fig. 4i–l; Fig. 4i–k). While SOX10+ cells are found within these tumors, they are sparse

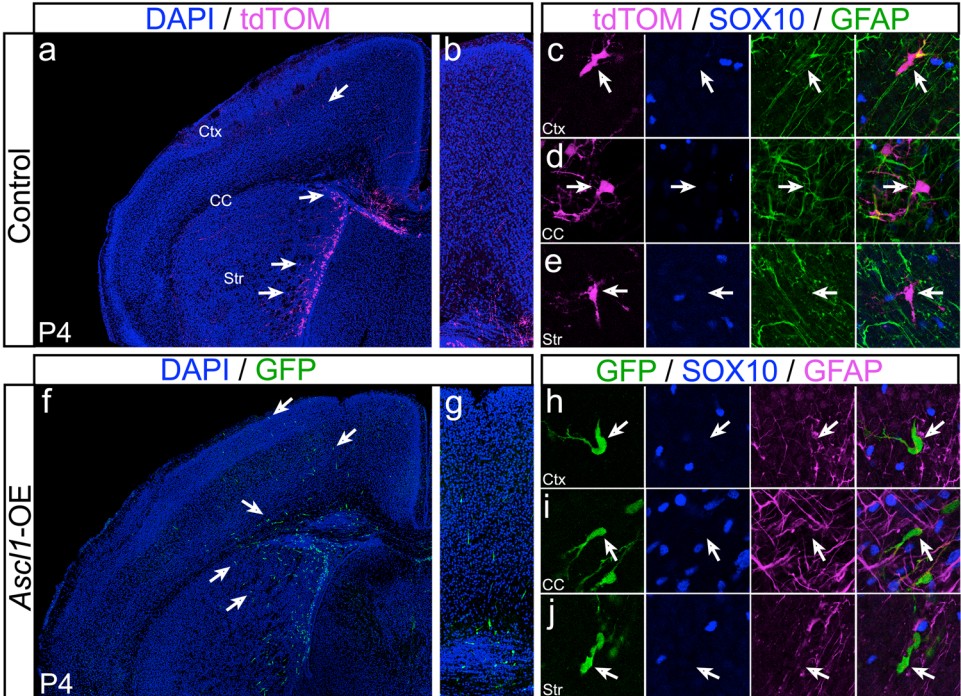

**Fig. 5 | ASCL1 overexpression promotes migration of newly transformed tumor cells. a–j** Immunofluorescence of SOX10 and GFAP in P4 brains of control (**a–e**) and *Ascl1*-OE (**f–j**) tumor mice. tdTOM+ cells of control brain are mostly restricted to the SVZ but GFP+ cells of *Ascl1*-OE brain have migrated extensively into the striatum (Str), corpus callosum CC, and cortex (Ctx) (arrows in (**a** vs **f**)). Note that neither tdTOM nor GFP co-localize with SOX10 or GFAP (arrows in (**c–e**) and (**h–j**)), indicating that they are unspecified migrating NPC-like tumor cells. Results were observed for *n* = 3 mice/genotype. Scale bar: 500 μm for (**a**, **f**); 100 μm for (**b**, **g**); 50 μm for (**c–e**, **h–j**).

and the majority did not co-localize with tdTOM (Fig. 4i–k; Supplementary Fig. 4u). Similarly, dCKO tdTOM+ tumors are predominantly GFAP+, and very few, if any, were SOX10+ (Supplementary Fig. 4m–p). However, the immunofluorescent level of GFAP appeared much lower in dCKO than in *Olig2*-CKO tumors (Fig. 4i–k vs m–o), likely due to the presence of ASCL1 in the latter. Finally, *Ascl1*-OE tumors, despite co-expressing OLIG2 (Fig. 3e, Supplementary Fig. 2k), exhibited high levels of GFAP and a significantly reduced number of SOX10+ cells (Supplementary Fig. 4q–u). Similar to control tumors, *Ascl1*-OE GFP+ tumor cells co-localized with either GFAP or SOX10 (Fig. 4q–s), indicating a "mixed glioma" phenotype. This highlights a discrepancy in SOX10+ versus OLIG2+ tumor cells, especially considering that *Sox10* is transcriptionally regulated by OLIG2 within OPCs[40]. This discrepancy was also observed within control tumors but in the opposite direction, whereby many SOX10+ tumor cells are negative for OLIG2 (Supplementary Fig. 4v,w).

The glioma phenotypes of these various tumors were also confirmed by H&E staining revealing the predominance of pink GFAP+ fibers (arrows) within astrocytoma tumors (*Olig2*-CKO, *Ascl1*-OE) or round "fried egg" cells with perinuclear halo (arrowhead), characteristics of oligodendroglioma (*Ascl1*-CKO) tumors (Fig. 4d, h, l, p, t). We also analyzed the expression of markers SOX2 and PDGFRA for all tumor types. Normally, SOX2 is expressed at high levels in NSCs/NPCs but at lower levels in astrocytes and OPCs[41,42], whereas PDGFRA is specific to OPCs[43]. We observed that tumor cells, irrespective of genotype, were SOX2+ (Supplementary Fig. 5). In contrast, PDGFRA was strongly detected in OLIG2+ tumors (control, *Ascl1*-CKO, *Ascl1*-OE) (Supplementary Fig. 5b, f, r), but was absent or greatly reduced in tumors without OLIG2 (*Olig2*-CKO and dCKO) (Supplementary Fig. 5j, n).

We next sought to determine if the highly diffuse nature of *Olig2*-CKO and *Ascl1*-OE tumors is due to the predominance of their astrocyte cell type or due to the presence/levels of ASCL1. To address this,

we compared the cell type and migration of newly induced control versus *Ascl1*-OE tumor cells at P4 (Fig. 5). At this stage, the majority of tdTOM+ or GFP+ cells have yet to express OLIG2, SOX10, or GFAP, as seen in P30 or terminal tumors, indicating that the glial fate of these cells is similar and has yet to be determined. While control tdTOM+ newly transformed tumor cells were mostly found adjacent to the SVZ and in the corpus callosum (arrows, Fig. 5a–e), *Ascl1*-OE GFP+ newly transformed tumor cells had migrated considerably into the striatum and upper cortical layers (arrows, Fig. 5f–j). This finding supports a direct role for ASCL1 in promoting tumor cell migration independent of its role in cell type specification.

## Tumor cells of the GBM mouse model are highly heterogeneous and contain all GBM subtypes

To better determine the role of ASCL1 in regulating the hierarchy and heterogeneity of tumor cells in the GBM mouse model, we next performed scRNA-seq analyses of control and *Ascl1*-OE tumors (Fig. 6a). These two tumor types were specifically chosen because they contained "mixed glioma" cell types, co-expressed both ASCL1 and OLIG2, but exhibited completely different migratory behavior. To avoid possible confounding transcriptomes from non-tumor cells (i.e., reactive astrocytes, microglia/macrophages, neurons), scRNA-seq was performed only on FAC-sorted tdTOM+ cells of control tumors (*N* = 3, 18,163 cells) or GFP+ cells of *Ascl1*-OE tumors (*N* = 3, 28,109 cells) (Fig. 6b; Supplementary Fig. 6). We detected a similar number of genes per cell for both tumor types (Fig. 6c). Unsupervised clustering using Uniformed Manifold Approximation and Projection (UMAP) analyses showed 16 transcriptionally diverse cell clusters across both tumor types, with cluster #5 completely segregated from the rest of the cell clusters (Fig. 6d–f). Gene expression analyses across these cell clusters confirmed that *Ascl1/GFP*, along with ASCL1 canonical NOTCH signaling target genes (*Dll3, Notch1, Hes5*) and the bHLH E-protein co-binding partners (*Tcf4, Tcf12*) were

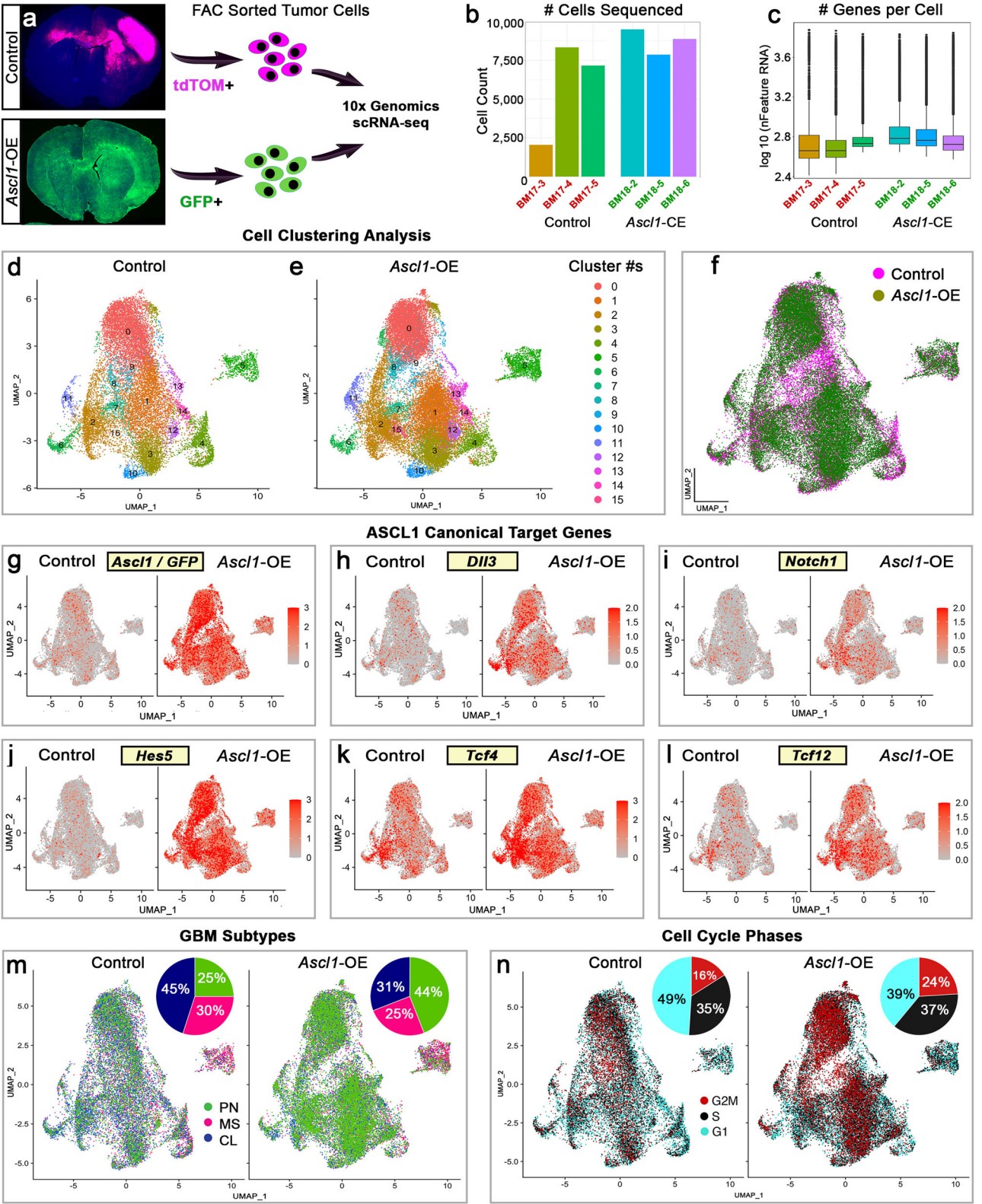

**Fig. 6 | Single-cell RNA-seq reveals high degrees of transcriptional diversity for control and *Ascl1*-OE tumor cells. a** Workflow of FAC-sorted tdTOM+ or GFP+ tumor cells for scRNA-seq. **b**, **c** Number of cells per tumor and genes per cell sequenced for control (*n* = 3) and *Ascl1*-OE (*n* = 3) tumors. Box plot extends from the 25th to the 75th percentile of each sample's distribution with the center line denoting the median number of genes per cell. Whiskers extend to 1.5 of the interquartile range. Dots are cells outside of the range. **d**–**f** UMAP visualization of unsupervised clustering of control and *Ascl1*-OE tumor cells yielded 16 different cell clusters. **g**–**l** Gene expression confirming increased levels of *Ascl1-ires-GFP*, known target genes *Dll3*, *Notch1*, and *Hes5*, as well as binding partners *Tcf4* and *Tcf12* in *Ascl1*-OE tumor cells compared to controls. **m**, **n** GBM subtypes (PN proneural, MS mesenchymal, CL classical) and cell cycle phase analyses of control and *Ascl1*-OE tumor cells.

significantly elevated in *Ascl1*-OE tumor cells compared to control (Fig. 6g–l).

We next assigned GBM subtype identities to all tumor cells using signature genes for proneural, classical, or mesenchymal GBM subtype[12,13]. All three subtypes were represented within both tumor models, with the proneural subtype expanded almost 2-fold (44%) in *Ascl1*-OE tumors at the expense of the classical subtype (31%) compared to control tumors (25% proneural, 45% classical), while the mesenchymal subtype was mostly unaffected (Fig. 6m). Cells in G2M phase were also increased in *Ascl1*-OE tumors (24%) compared to control tumors (16%) (Fig. 6n). These results are consistent with ASCL1's function as a proneural factor[12] and role in promoting tumor cell proliferation[20] (Supplementary Fig. 3k, l).

## High levels of ASCL1 promote NSC/astrocyte-like cell types at the expense of OPC/oligodendrocyte-like cell types in brain tumors

We next analyzed how the lineage composition of tumor cells was altered in *Ascl1*-OE versus control tumors. We assigned the tumor cells a predicted cell type based on cell type-specific signature genes curated from previously published data sets. These data sets included scRNA-seq of quiescent and active NSCs (qNSC, aNSC) isolated from the adult SVZ[44–46], and both scRNA-seq and cell type-specific purified bulk RNA-seq of astrocyte, oligodendrocyte (OPC, NFOL, MOL), and microglial lineage cells isolated from juvenile and adult mouse brains[47–50]. We selected about 40-50 signature genes per cell type (Supplementary Data 7), each of which was validated to be highly expressed and specific to their respective cell type using BrainRNA-seq.org, especially for astrocyte, OPC, NFOL, MOL, and microglia. These signature genes were also cross-referenced with our scRNA-seq data to ensure that they were sufficiently expressed in tumor cells of our GBM mouse model.

In both control and *Ascl1*-OE tumors, approximately 90% of the tumor cells were assigned a CNS cell type (OPC, NFOL, MOL, qNSC, aNSC, or astrocyte), while the remaining 10% were assigned a microglia cell type, the majority of which are found in cluster #5. Notably, clusters # 12, 14, and 4 are assigned astrocyte, NFOL, and MOL cell types, respectively, while the remaining twelve clusters (dotted line) contained all seven assigned cell types, though at varying proportions and predominated by OPCs, NSCs, and astrocytes (Fig. 7a–f). For control tumor cells, about 74% were assigned oligodendrocyte lineage cell types (40% OPC, 17% NFOL, 17% MOL) (Fig. 7a), 19% were of NSC and astrocyte cell types (8% qNSC, 7% aNSC, 4% AS) (Fig. 7b), and the remaining were assigned as microglia (7%) (Fig. 7c). Compared to control, *Ascl1*-OE tumor cells showed a 3-fold decrease in oligodendrocyte lineage cell types, with OPCs (4%) decreasing by 10-fold while both NFOL (11%) and MOL (11%) each showed a 6% decrease (Fig. 7d). These decreases were accompanied by a 6-fold increase in astrocyte (25%) and a 2- to 3-fold increase in qNSC (17%) and aNSC (21%) assigned cell types (Fig. 7e). The proportion of microglia (11%) was slightly increased by *Ascl1*-OE (Fig. 7f). Accordingly, the switch in assigned cell types in *Ascl1*-OE tumors was associated with specific downregulation of OPC, NFOL, and MOL signature genes and upregulation of NSC and astrocyte signature genes (Fig. 7g, h).

We then combined the transcriptomes of each assigned cell type into Unionized Cell Type RNA-seq (Supplementary Data 8) for each tumor to better demonstrate the specificity of the cell type signature genes and how their expression was altered in the triplicates of *Ascl1*-OE tumors (BM18-2, BM18-5, BM18-6) compared to control tumors (BM17-3, BM17-4, BM17-5). Heatmap analysis showed that expression of the signature genes was specific to their respective assigned cell types (black rectangles, Fig. 7i). Interestingly, within control tumors, many OPC and some NSC genes were also expressed across other assigned cell types, indicating that the more differentiated tumor cell types are likely derived from these earlier progenitor/stem-like cells.

As seen for the scRNA-seq (Fig. 7g, h), oligodendrocyte lineage (OPC, NFOL, MOL) signature genes were significantly downregulated, especially OPC signature genes, as seen across all cell types in *Ascl1*-OE tumors (blue arrows, Fig. 7i). On the other hand, NSC and astrocyte signature genes, especially those expressed at low levels in control tumors, were markedly upregulated in all cell types of *Ascl1*-OE tumors (red arrows, Fig. 7i), while microglia signature genes were largely unaffected. This pattern of differential regulation was also observed when displayed visually across UMAP cell clusters, even though many OPC/oligodendroglial (*Olig2, Sox10, Pdgfra, Ctntn1, Chn2, Mobp, Mbp*), NSC/astroglial (*Sox2, Sox4, Gfap, Birc5, Egfr, Id3, Aqp4*), and microglial (*Tmem119, Laptm5, Selplg, Cst7, Hexb, Ctsb, Ly86*) signature genes are targets of ASCL1 binding (Fig. 8a–c; TS1).

To validate our findings with that of human GBMs, we next used signature gene modules of GBM cancer cell states (NPC-like, AC-like, OPC-like, and MES-like) previously reported by Neftel et al. (2019)[16] to assign cell states to our control and *Ascl1*-OE mouse tumor cells. Consistent with our finding, relative meta-module score analysis (threshold > ± 0.5) revealed that NPC-like and AC-like cell states were over-represented in *Ascl1*-OE tumors at the expense of the OPC-like cell state in comparison to control tumors, while the MES-like cell state was unaffected (Supplementary Fig. 7a, b). Accordingly, heatmap analysis of Unionized Cell-Type RNA-seq also demonstrated extensive downregulation of OPC-like genes and upregulation of NPC-like and AC-like genes in *Ascl1*-OE tumors (Supplementary Fig. 7c–h).

## Ribosomal protein, mitochondrial, cancer metastasis, and stem cell maintenance genes are upregulated in NSC/astrocyte-like cells of *Ascl1*-OE tumors

To identify genes (whether directly or indirectly regulated by ASCL1) that may contribute to the highly invasive and aggressive phenotype of *Ascl1*-OE tumors, we first identified differentially expressed genes (DEGs, Log$_2$FC ≥ ± 0.33) compared to control tumors for each assigned cell type. Low-expressing genes in less than 20% of cells in any cell type in both control and *Ascl1*-OE tumors were excluded to ensure that only highly expressed genes were considered. This resulted in 421 upregulated and 197 downregulated DEGs, which included both ASCL1 target and non-target genes (Fig. 9a; Supplementary Data 9). GO analysis of the upregulated DEGs showed that the top genes and biological processes highly enriched were involved in translation and mitochondrial inner membrane, which are essential for sustaining the necessary protein biosynthesis and metabolic support required for highly proliferative and migratory cancer cells[51] (Fig. 9b; Supplementary Data 10). The upregulation of ribosomal protein large and small (*Rpl/Rps*) subunit encoding genes important for translation was particularly high within the microglial tumor cell types, while oxidative phosphorylation genes important for ATP production were upregulated across all assigned cell types (Fig. 9c–e).

We next focused on the top 25 up- or downregulated DEGs (Log$_2$FC ≥ ± 1) that were observed in at least 30% of cells of five or more of the seven cell types in *Ascl1*-OE and/or control tumors. Not surprisingly, an overwhelming number of the top 25 upregulated DEGs are known to promote cancer metastasis and invasion (*Crip1, Gfap, Sparcl1, Tmsb4x, Apoe, Ier2, Sparc, Cpe, Fabp5, Cdk2ap1, Jpt1, Vim, Ckb*)[52–64], NSC/GSC maintenance (*Id1, Id3, Hes5, Gfap, Lgals1, Sparcl1, Tmsb4x, Fos*)[54,65–71], and/or resistance to chemotherapy (*Crip1, Gfap, Apoe, Mt1, Mt3, Lgals1*)[64,69,72–74] (Fig. 9f). Some of these top-upregulated DEGs are also the most highly expressed genes (*Apoe, Clu, Cpe, Cst3, Sparcl1, Ckb, Slc1a2, Mt1, Mt3*) in cortical astrocytes in the postnatal brain[47]. In contrast, the top 25 DEGs that were downregulated are known to result in suppression of inflammatory and immune responses (*U2af1l4, Psenen, Tbcb, Tmem147, Yif1b, S100b*)[75–80], prevent NSC differentiation (*Clip3*)[81], or promote cancer metastases (*Tbcb, Usf2, Tpm1*)[82–84] and resistance to drug or stress induced apoptosis (*Aplp1, Rbm42, Capns1, Eif3k, Tpm1*)[85–89] (Fig. 9g). Three of the top upregulated

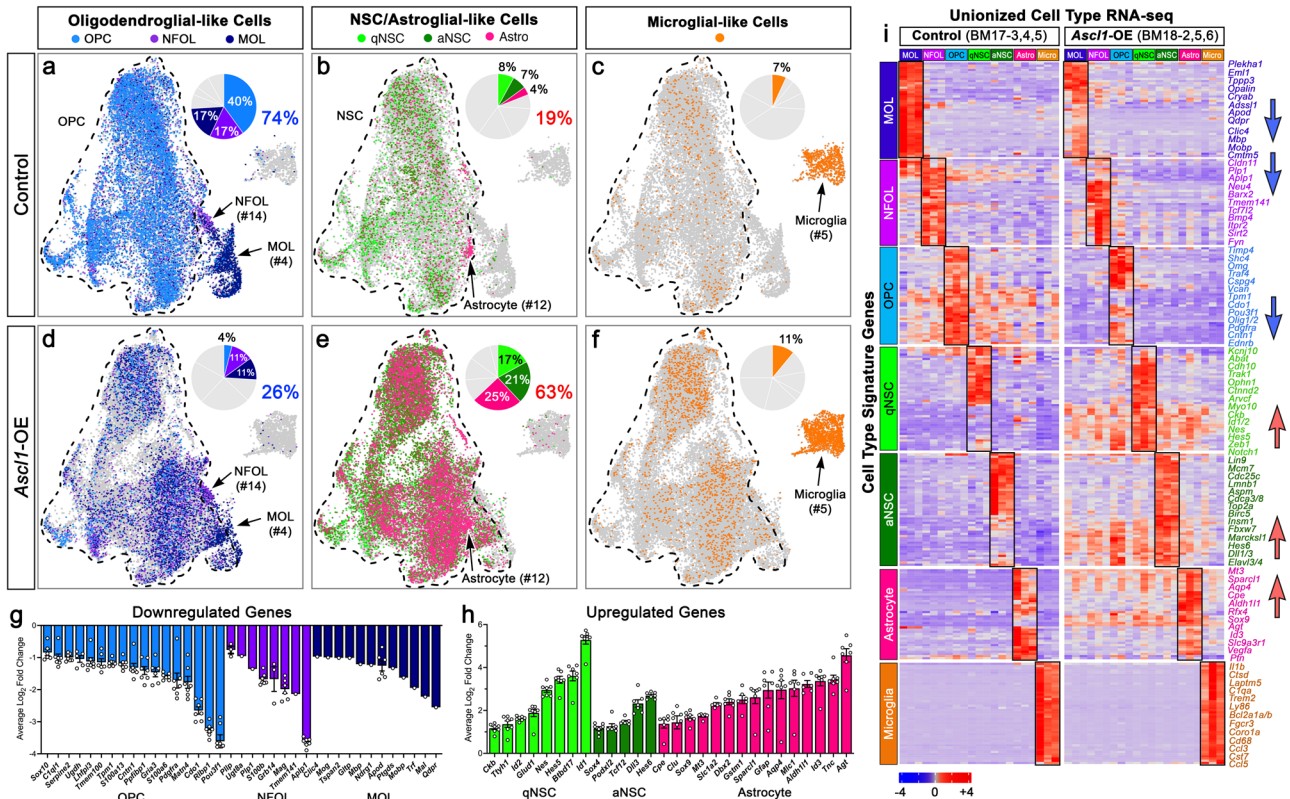

**Fig. 7 | *Ascl1* overexpression promotes NSC/astroglial-like cells and suppresses OPC/oligodendroglial-like cells in mouse GBMs. a–f** UMAPs demonstrating the proportion and distribution of 7 assigned cell types in control (**a**–**c**) and *Ascl1*-OE (**d**–**f**) tumors based on cell type-specific gene signatures. **g, h** Differential gene expression confirms downregulation of oligodendrocyte lineage-specific genes (**g**) and upregulation of NSC/astrocyte-specific genes (**h**) in *Ascl1*-OE tumor cells. Bar graphs are mean Log$_2$ Fold Change ± SEM for indicated genes by comparing cell type x cell type between control and *Ascl1*-OE tumors. Open circles within each bar graph represent number of cell types with indicated genes significantly altered

(adjusted *p*-value < 0.05). Note that MOL genes are mostly restricted to MOL and thus only downregulated in that cell type. **i** Heatmap of averaged transcriptome for each assigned cell type into unionized RNA-seq triplicates (columns) for control and *Ascl1*-OE tumors showing specificity of signature genes (rows) to their respective cell types (black rectangles). Note that all cell types of control tumors expressed some level of OPC signature genes, which were the most drastically downregulated in *Ascl1*-OE tumors followed by NFOL and MOL signature genes (blue arrows), while NSC and astrocyte signature genes were highly upregulated across all cell types in *Ascl1*-OE tumors (red arrows).

and most highly expressed DEGs in the majority of *Ascl1*-OE tumor cells are *Tmsb4x, Sparcl1,* and *Mt3* (Fig. 9h). Conversely, *Psenen, Tbcb,* and *Tmem147* are three of the top downregulated DEGs that were highly expressed in control tumor cells but were drastically reduced across *Ascl1*-OE tumor cells (Fig. 9i).

Collectively, these findings demonstrate a pivotal role for ASCL1 as a master regulator of genes essential to sustain the highly proliferative, migratory, and therapeutic-resistant potential of astrocyte-like GSCs within GBM tumors in the brain.

## Discussion

Defining hallmarks of GBM include high degrees of inter- and intra-tumoral cellular and molecular heterogeneity combined with rapid proliferation and invasion of the brain. Studies of both human and genetically engineered mouse models implicate genomic alterations, driver mutations, cell-of-origin, and/or the tumor microenvironment as potential determining factors of the heterogeneity, lineage hierarchy, and aggressiveness of GBMs[12,13,90–93]. However, precisely how these factors are translated within tumor cells to generate a variety of molecularly and behaviorally distinct GBM cellular subtypes remains unclear.

During gliogenesis in the developing cortex, the expression of ASCL1 promotes the generation of intermediate NPCs from radial glia[22]. The sustained expression of ASCL1 then leads to direct transcriptional activation of *Olig2* and NOTCH genes (*Dll1, Dll3, Hes5, Notch1*) to specify the fate of NPCs as they migrate out of the

ventricular zone into the cortical plate[22,94–96]. Notably, NPCs with sustained high levels of OLIG2 are specified into OPCs (SOX10+, PDGFRA+), whereas those with lower levels of OLIG2 and higher NOTCH signaling are specified into astrocyte precursor cells (APCs) (EGFR+, HES5+, ID3+, NFIA+). Although ASCL1 is maintained and required for the proliferation of postnatal OPCs[21], it is downregulated as both APCs and OPCs differentiate into mature astrocytes (GFAP+, ALDH1L1+, AQP4+) and oligodendrocytes (CC1+, MAG+, MBP+), respectively (Fig. 10a)[21,22].

In this study, we showed that induction of oncogenic driver mutations in radial glia in the SVZ leads to the dysregulation of ASCL1 and OLIG2. This dysregulation likely occurs at both the transcriptional and protein levels, wherein the expression of ASCL1 and/or OLIG2 further reciprocally and positively sustain each other's expression (Supplementary Fig. 3). In parallel, ASCL1 and OLIG2 activity may also be enhanced by serine-threonine kinase (i.e., MAPK, ERK) driven phosphorylation, resulting in stabilization of these transcription factors to selectively promote activation of cell cycle, NOTCH, and NSC programs[97–101]. Consequently, affected radial glia are then transformed into tumor-propagating NPCs marked by dynamic levels of ASCL1 and OLIG2. Similar to during gliogenesis, the levels and functional interactions of ASCL1 and OLIG2 then determine the cell types and degree of migration of glioma tumors that are generated (Fig. 10b). Loss- and gain-of-function combined with scRNA-seq reveal that tumor cells with higher levels of ASCL1 relative to OLIG2, as seen in the *Olig2*-CKO and *Ascl1*-OE tumors, favor activation of a NSC/astrocyte program,

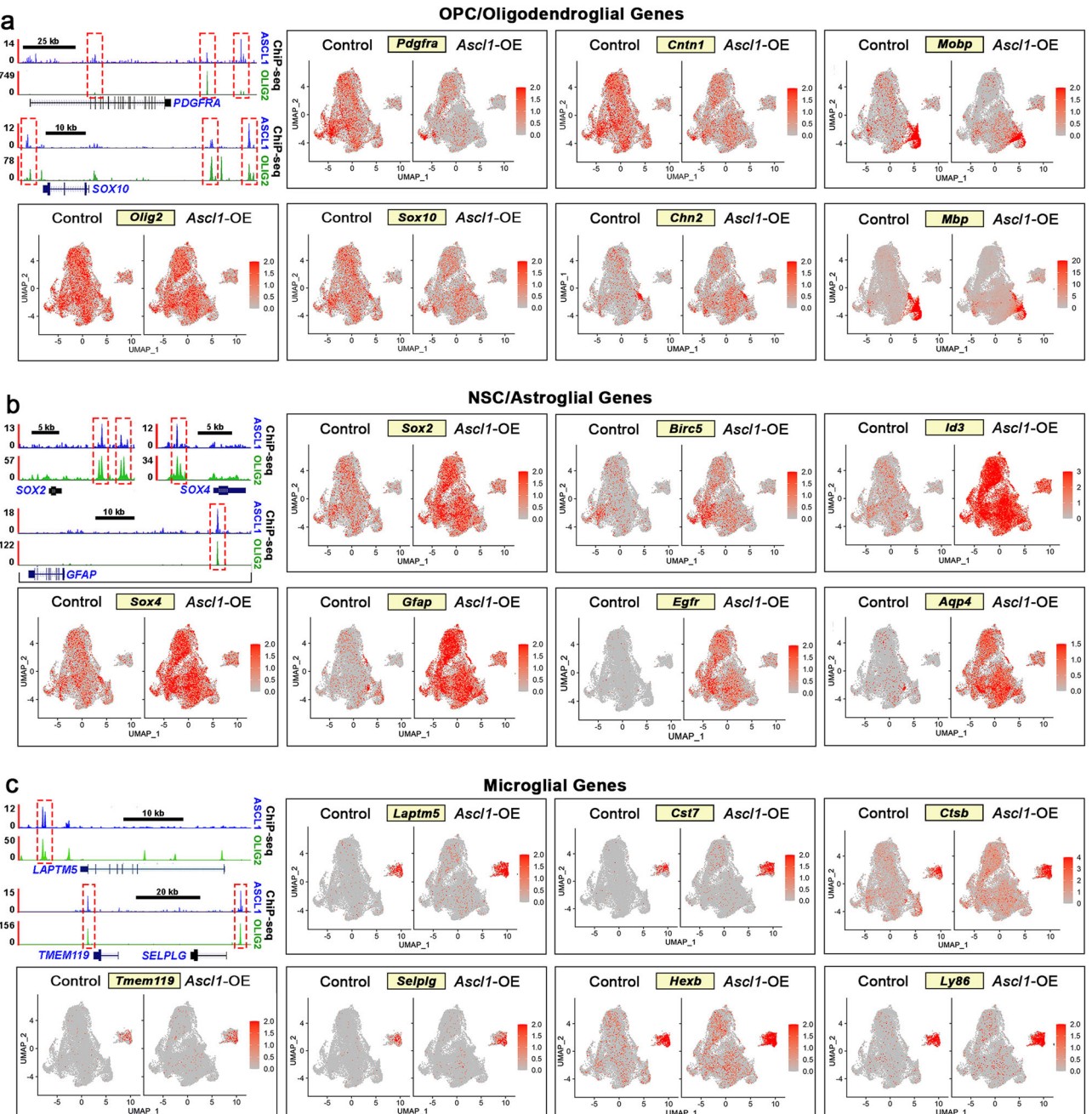

**Fig. 8 | ASCL1 and OLIG2 shared target cell type-specific genes are differentially expressed in control and *Ascl1*-OE tumors. a–c** ChIP-seq tracks demonstrating shared binding of ASCL1 and OLIG2 at cis-regulatory sites of OPC/oligodendroglial (**a**), NSC/astroglial (**b**), and microglial lineage genes (**c**) and UMAP showing differential expression in control and *Ascl1*-OE cell clusters. All genes illustrated are shared targets of ASCL1 and OLIG2.

characterized by high expression of GFAP and a highly migratory or diffuse phenotype (Fig. 10c, d). These ASCL1^high cells, which showed an upregulation of NSC maintenance, cancer metastasis, and therapeutic resistance genes, are likely the GPCs or GSCs previously characterized in human GBM tumors[30]. In contrast, tumor cells with higher levels of OLIG2 relative to ASCL1, such as *Ascl1*-CKO tumors, favor activation of an OPC/oligodendrocyte program and a less diffuse tumor phenotype (Fig. 10d). Finally, glioma initiation and formation were completely compromised in the majority of *Ascl1;Olig2*-dCKO mice (Fig. 10e), highlighting the redundant function of ASCL1 and OLIG2 as prominent transcription factors hijacked by tumor-initiating events.

Our ChIP-seq and co-immunoprecipitation assays for ASCL1 and OLIG2 in two orthotopic lines of GBMs reveal critical insights into the

shared and intricate binding of these two bHLH transcription factors. Notably, although OLIG2 binds more genomic sites than ASCL1, there is extensive overlap with ASCL1 binding sites (Fig. 1a). This binding overlap may be due to the direct dimerization between ASCL1 and OLIG2 (Fig. 3j). Target genes associated with ASCL1 and OLIG2 shared binding sites are comprised of a complex transcriptional network essential for sustaining GBM cells in a persistent state of undifferentiation, proliferation, and malignancy. Major target genes of this transcriptional network include *ASCL1* and *OLIG1/2* themselves, their respective bHLH E-protein co-binding partners (*TCF3, TCF4, TCF12*), NOTCH signaling (*DLL1, DLL3, NOTCH1, HES5, HES6*), and a multitude of cell cycle, NSC, astrocyte, and oligodendrocyte lineage genes (Fig. 1; Supplementary Data 2). The significance of this transcriptional

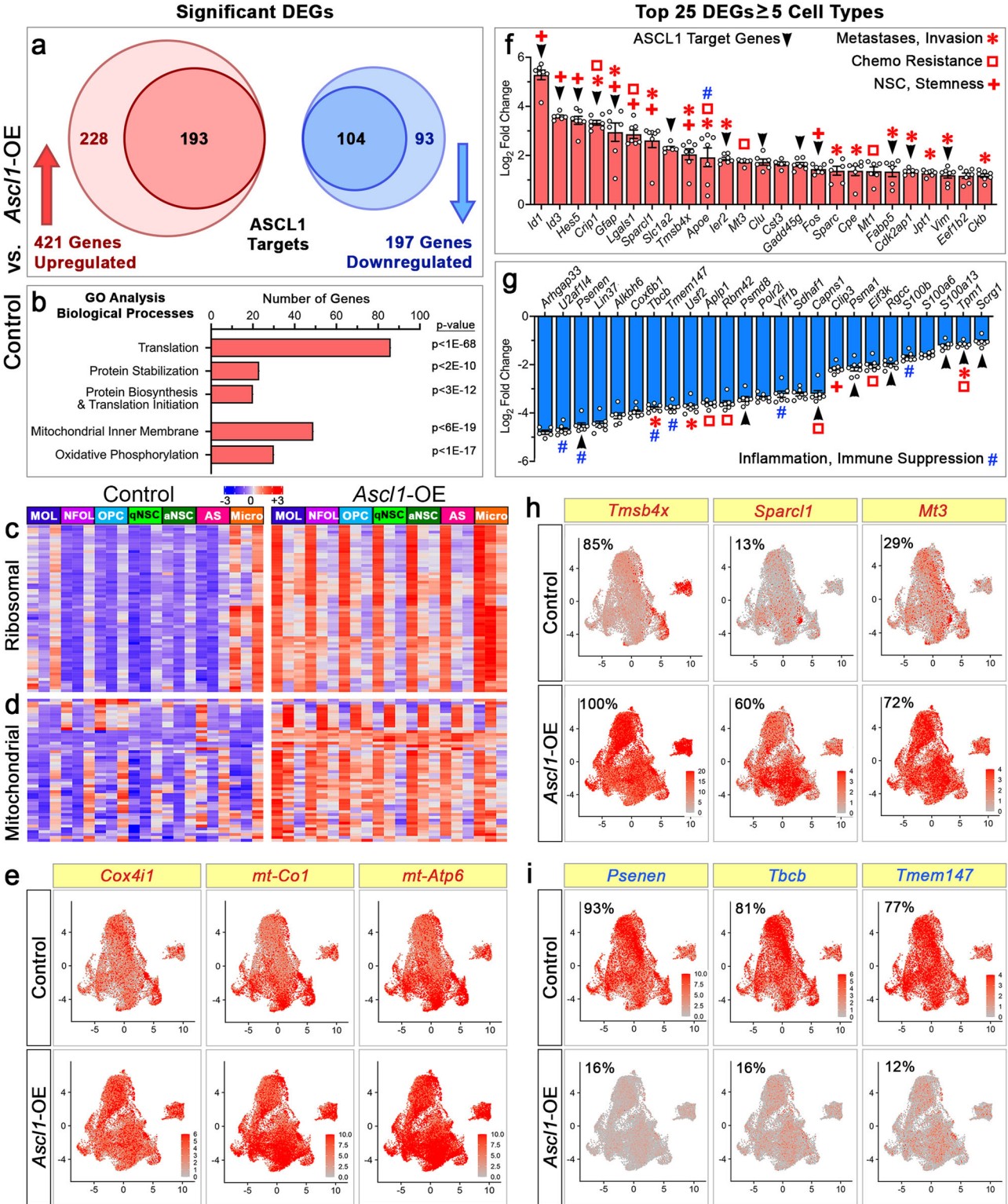

**Fig. 9 | Genes highly upregulated in *Ascl1*-OE tumors are important for NSC maintenance, cancer metastasis and invasion, and therapeutic resistance.**
**a** Differentially expressed genes (DEGs) significantly up- or downregulated by comparing cell type x cell type between control and *Ascl1*-OE tumors. DEGs include both ASCL1 target and non-target genes. **b** Gene ontology analysis of upregulated DEGs showing enrichment of genes important for protein synthesis and mitochondrial function. **c, d** Heatmap of Unionized Cell-Type RNA-seq (columns) demonstrating upregulation of ribosomal (**c**) and mitochondrial (**d**) genes (rows) in *Ascl1*-OE tumors. **e** UMAPs showing specific upregulation of cytochrome C oxidase

subunit genes. **f, g** Top 25 genes upregulated (**f**) or downregulated (**g**) in ≥ 5 cell types, with delineation of ASCL1 targets and known functions in cancer or GBMs. Bar graphs are mean Log$_2$ Fold Change ± SEM for indicated genes by comparing cell type x cell type between control and *Ascl1*-OE tumors. Open circles within each bar graph represent number of cell types with indicated genes significantly altered (adjusted *p*-value < 0.05). UMAP gene expression of 3 of the topmost upregulated (**h**) and downregulated (**i**) genes. Proportions of tumor cells expressing these genes are indicated for control and *Ascl1*-OE tumors.

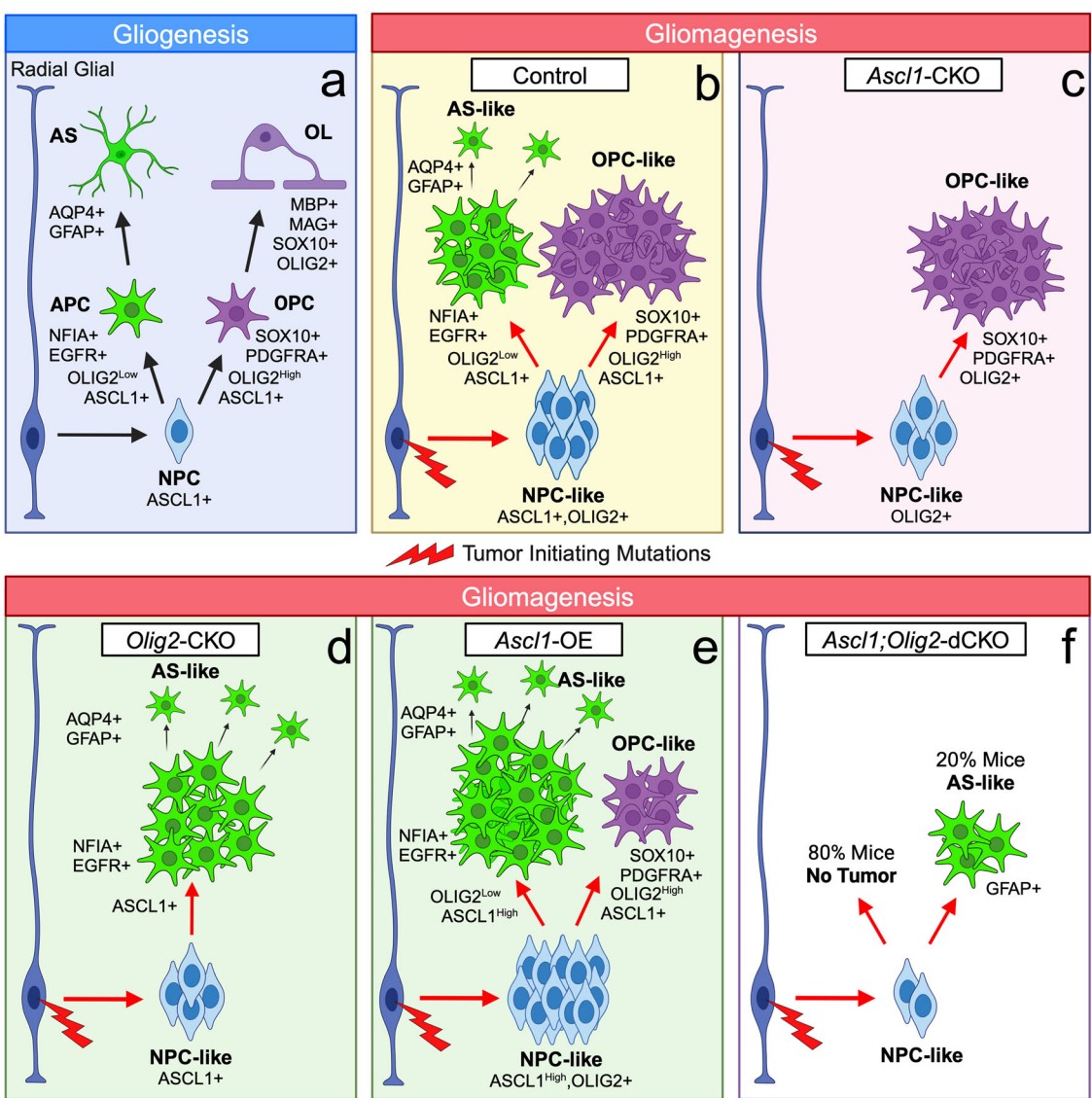

**Fig. 10 | Model of ASCL1 & OLIG2 Function in Gliogenesis and Gliomagenesis.**
**a** Schematic of the role of ASCL1 and OLIG2 in glial cell fate specification in the dorsal forebrain. Developmentally, ASCL1 expression leads to generation of intermediate NPCs from radial glia and transcriptional activation of *Olig2*. Depending on sustained levels of OLIG2, NPCs are specified into glial precursor cells (APCs or OPCs). Markers of astrocyte and oligodendrocyte lineages are indicated.
**b**–**f** Schematic summary of tumor induction from radial glia and the role of ASCL1

and OLIG2 in specifying glioma cell types in control (**b**), *Ascl1*-CKO (**c**), *Olig2*-CKO (**d**), *Ascl1*-OE (**e**), and *Ascl1;Olig2*-dCKO (**f**) tumors. High levels of ASCL1 specifies NPC-like and AS-like tumor cells, which are highly migratory, whereas high levels of OLIG2 specifies OPC-like tumor cells. Tumor induction is mostly compromised in the absence of both ASCL1 and OLIG2. Created in BioRender. Myers, B. (2024) BioRender.com/p53n344.

network in gliomagenesis was confirmed by their correlated expression with both *ASCL1* and *OLIG2* in RNA-seq of TCGA GBM samples (Supplementary Data 3, 4). More importantly, we also validated in the various mouse brain tumor types that genes of this transcriptional network and the cell types (NSC/NPC, astrocyte, OPC) associated with their function were altered accordingly in the absence or elevated levels of ASCL1 and/or OLIG2 (Figs. 4–9).

Although tumor cells induced from the SVZ of this GBM mouse model exhibit similar glial cell fate specification as observed during gliogenesis, there were some fundamental differences within the context of gliomagenesis. Notably, terminal control tumors exhibited a much higher percentage of SOX10+ cells (~90%) than OLIG2+ cells (61%), whereby SOX10 was observed in a subset of OLIG2-;tdTOM+ tumor cells (Supplementary Fig. 4v, w). This aberrant discrepancy was also observed in *Olig2*-CKO and dCKO tdTOM+ astrocytic tumors, in which about 10% of tumor cells are SOX10+ (Supplementary Fig. 4u).

This expression of SOX10 in the absence of OLIG2 does not occur during glial development[40] and may reflect the aberrant dysregulation of the glioma genomic landscape, which is consistent with previous work showing time-specific fate-switching of glioma cells to favor oligodendrocyte-like cells at end stages of tumor progression[100]. Additionally, although *Ascl1*-OE tumors highly co-expressed ASCL1 and OLIG2 (Fig. 3e; Supplementary Fig. 2k), both of which are required for the specification and development of OPCs[21,23,24,102,103], NPC/astrocyte-like cells seem to make up the majority of the cells within these tumors rather than OPC-like cells. Moreover, despite the shared binding of ASCL1 and OLIG2 to signature genes of both NSC/astroglial and OPC/oligodendroglial lineages, scRNA-seq of *Ascl1*-OE tumor cells revealed an upregulation of NSC/astrocyte signature genes but a down-regulation of OPC/oligodendrocyte lineage genes compared to control tumors (Figs. 7, 8). A possible explanation for this finding is that similar to that observed in NSCs[104], a sustained high level of ASCL1 leads to an

upregulation of *Hes5* (Fig. 6j), which in turn downregulates *Olig2*[27,28], as revealed at both the transcript and protein levels in *Ascl1*-OE compared to control and *Ascl1*-CKO tumor cells (Figs. 3g, 8a). Consequently, this downregulation of OLIG2 is not optimal to sufficiently induce expression of oligodendrocyte lineage markers, as highlighted by the decrease in SOX10+ cells within *Ascl1*-OE tumors (Supplementary Fig. 4u). Alternatively, the decrease in SOX10+ cells may also be due to the upregulation of *Nfia* (Supplementary Data 9), a target of ASCL1 that has been shown to mutually antagonize the induction of SOX10, thus suppressing OPC/oligodendrocyte fate in favor of an NSC/astrocyte fate within glioma tumors[105]. Previously, the RNA-binding zinc-finger protein-encoding gene, *Zfp36l1*, was also shown to control oligodendrocyte-astrocyte fate transition, whereby conditional knock-out results in downregulation of oligodendrocyte lineage genes in favor of upregulation of astrocyte-lineage genes in both the developing forebrain and tumors of a glioma mouse model[93]. Surprisingly, *Zfp36l1* is upregulated in *Ascl1*-OE tumor cells, especially in qNSC and aNSC assigned cell types (Supplementary Data 9), likely because it is a target of ASCL1 (Supplementary Data 2). This upregulation suggests that *Zfp36l1* may contribute to the maintenance or specification of these cell types, and its function may be context or cell-type dependent since it is expressed in NPCs and astrocytes in addition to oligodendrocyte lineage cells[93].

Of significant importance to the lineage hierarchy and heterogeneity of GBMs is that the *Ascl1*-OE phenotype, combined with the *Olig2*-CKO phenotype, offers two important insights into the functional interactions between ASCL1 and OLIG2 and the significance of their roles in regulating opposing NSC/astrocyte-like versus OPC/oligodendrocyte-like cell types in brain tumors, respectively. The first revelation is that without OLIG2, ASCL1 cannot efficiently activate the transcription of oligodendrocyte lineage genes, indicating that ASCL1's binding to these genes in the context of gliomagenesis, and possibly gliogenesis, may require a direct dimerization with OLIG2. In contrast, OLIG2's binding of oligodendrocyte lineage genes can occur in the absence of ASCL1, as seen in *Ascl1*-CKO tumors (Fig. 4e–h). Secondly, OLIG2 may suppress NSC/astrocyte fate in part through the direct dimerization with ASCL1, likely by recruiting it to preferentially bind to OPC/oligodendrocyte over NSC/astrocyte lineage genes. Thus, in the absence (*Olig2*-CKO) or presence of lower levels of OLIG2 (*Ascl1*-OE) within tumor cells, ASCL1 is able to escape this dimerization or repression to potently activate NSC/astrocyte genes (Fig. 4i–l, q–t). The marked increase of GFAP in both *Ascl1*-OE and *Olig2*-CKO tumors compared to their respective control and dCKO tumors suggests that this astrocytic phenotype is dependent on ASCL1 (Fig. 4). However, the functional impact of ASCL1's function in the context of GBM extends beyond just cell type/subtype commitment and tumor cell proliferation but also to produce highly migratory and diffuse tumor cells. This was revealed by ASCL1's ability to promote an increase in early migration of newly transformed GFP+ tumor cells out of the SVZ at P4 into the striatum, corpus callosum, and cortical plate of *Ascl1*-OE mice prior to their specification into either GFAP+ or SOX10+ tumor cells (Fig. 5). Additionally, although *Olig2*-CKO tumors induced from adult OPCs in the cerebral white matter of a GBM mouse model exhibit a strong astrocytic phenotype, these tumors seem to be mostly non-diffuse[33]. This astrocytic phenotype is unlike the highly diffuse *Olig2*-CKO astrocytic tumors of this study generated from neonatal radial glia in the SVZ. This migration difference could be due in part to the manner of tumor induction and/or tumor cell-of-origin, or possibly due to the low levels of *Ascl1* in these adult OPC-induced *Olig2*-CKO tumors, as revealed by bulk RNA-seq[33]. Interestingly, while scRNA-seq of *Ascl1*-OE tumor cells demonstrates that some of the top most upregulated DEGs are associated with cancer metastases and invasion (*Crip1, Gfap, Sparcl1, Tmsb4x, Apoe, Ier2, Gadd45g, Sparc, Cpe, Fabp5, Cdk2ap1, Jpt1, Vim, Ckb*), and thus may contribute to the highly aggressive and diffused phenotype of these tumors, there was no significant change in *Rnd3* expression, a direct target of ASCL1 (Supplementary Data 2) that has been shown to be important for neuronal migration[36]. This implies that ASCL1's role in conferring tumor cell migration within GBMs may utilize downstream mechanisms independent of RND3 function.

Our scRNA-seq of tens of thousands of FAC-sorted control and *Ascl1*-OE tumor cells from the GBM mouse model provides an unprecedented view of the highly dynamic transcriptomic landscape of brain tumors relevant to our understanding of the biology, plasticity, and malignancy of human GBMs (Figs. 6–9). Notably, all GBM subtypes (proneural, classical, mesenchymal) were represented in all the mouse brain tumors, and as expected, the proneural subtype was expanded at the expense of the classical subtype with over-expression of ASCL1 due to its function as a proneural factor (Fig. 6j). Interestingly, this expansion of the proneural subtype was accompanied by an increase in NSC/astroglial-like cells and a decrease in OPC/oligodendroglial-like cells, which have traditionally been associated with proneural GBMs[12]. This unexpected change in GBM subtype with tumor cell type also holds when *Ascl1*-OE tumor cells were assigned using cancer cellular state signature genes derived from human GBMs[16], demonstrating that GBM subtype identities, especially proneural and classical, may be better defined by transcriptional targets of ASCL1 and/or OLIG2 rather than the glial cell types by which they may share some overlapping marker gene expression. Furthermore, we found that all CNS cell types (qNSC, aNSC, astrocyte, OPC, NFOL, MOL) and microglia/mesenchymal-like cells were also represented within the scRNA-seq of mouse brain tumors. UMAP clustering shows that except for a few cell clusters that are comprised exclusively of microglia, astrocytes, NFOL, and MOL, the majority of the tumor cells did not cluster together according to any assigned cell types. The dynamics or transient nature of these cell types are nonetheless consistent with scRNA-seq of primary GBMs in which NPCs or GPCs are situated at the apex of several transient cancer cell states consisting of OPC/oligodendroglial, astroglial, and mesenchymal cancer cells[16,30]. Notably, the NPC/GPC-like cancer cells are marked by ASCL1 or its direct target genes (*DLL3, HES5*) and are resistant to TMZ treatment[30]. In support of these human findings, we directly show that a high level of ASCL1 is necessary and sufficient to confer tumor cells of our GBM mouse model with highly proliferative, migratory, and likely therapeutic-resistant potential by directly and indirectly upregulating genes essential for cancer stemness, invasion, and chemoresistance. These upregulated genes include the vast majority of ribosomal protein small/large subunit encoding genes and mitochondrial oxidative phosphorylation genes (*Cox4i1*) critical for sustaining the costly translational and metabolic demands of CSCs[106–113] (Fig. 9). Notably, one of the top-most ubiquitously expressed and significantly upregulated genes that may directly contribute to the GSC phenotype of *Ascl1*-OE tumors is *Tmsb4x*, which encodes for Thymosin β−4, a potent regulator of actin polymerization[114]. Overexpression of *Tmsb4x* has been shown to positively regulate NPC expansion and confer stemness and chemotherapeutic resistance, while silencing promotes stem cell differentiation and decreases the invasion and proliferation of glioma cells[68,70]. In human gliomas, Thymosin β−4 levels are highest in GBMs and inversely correlated with survival[70], indicating that Thymosin β−4 may serve as a potential therapeutic target.

In summary, our comprehensive in vivo analyses of ASCL1 and OLIG2 loss- and gain-of-functions in primary brain tumors provide proof of concept of the combinatorial function of these two bHLH transcription factors in determining the lineage hierarchy, heterogeneity, and plasticity of GBMs. Furthermore, the robustness of our study in generating the various glioma tumor types in the brains of immunocompetent mice combined with our identification of genes that may directly or indirectly contribute to the highly aggressive and

invasive nature of GSCs offer exciting opportunities to further elucidate and mitigate the mechanism of therapeutic resistance in GBMs.

## Methods

### Mouse strains used for this study

All mouse experiments in this study followed NIH Guide for Care and Use of Laboratory Animals and in accordance with a research protocol approved by the Institutional Animal Care and Use Committee (IACUC) at the University of New Mexico Health Sciences Center and UT Southwestern. All mice were maintained in a 12 h light/dark cycle (lights off 1800 hours), temperature, and humidity-controlled facility with food and water available ad libitum. The generation and genotyping of mouse strains used in this study are described in Supplementary Table 1.

### ChIP-seq and TCGA RNA-seq data analyses

Details of our ChIP-seq assays for ASCL1 (GSE152401) using two orthotopic lines of patient-derived GBM xenograft (PDOX-GBMs) (R548 and R738) were previously reported[20]. Here, ChIP-seq assays for OLIG2 were done for the same two PDOX-GBM lines to directly compare the binding profiles of ASCL1 and OLIG2. Briefly, PDOX-GBMs were grown and dissected from brains of NOD-SCID mice and then homogenized and fixed in 1% formaldehyde to crosslink proteins and DNA, followed by quenching with 0.125 M of glycine. Nuclear chromatin was pelleted, washed with cold PBS, and sonicated into 200−300 bp fragments using a Biorupter (Diagenode). A portion of the sheared chromatin (10%) was set aside as input DNA, while 100 μg was subjected to chromatin immunoprecipitation (ChIP) using 5 μg of mouse anti-ASCL1 (Mash1) antibody (BD Biosciences, 556604) or 5 μg rabbit ant-OLIG2 (Olig2) (Millipore, AB9610). Washes and reverse-crosslinking were performed using Dynabeads Protein G to elute the ChIP DNA for sequencing.

ChIP-seq analyses were performed as we previously reported[20]. Briefly, bowtie2 (v2.2.6)[115] was used to align sequence reads to the human reference genome (hg19), and de novo motif analyses were performed using HOMER Software (v.4.7). Target genes associated with ASCL1 or OLIG2 ChIP-seq peaks were determined using GREAT v4.0.4 (http://great.stanford.edu/public/html/)[31]. We used Spearman rank order correlation (>0.4) to identify the top 10% of genes found in bulk RNA-seq of GBM samples in the TCGA cohort[11] positively correlated with *ASCL1* and/or *OLIG2* expression[20]. Biovenn (https://www.biovenn.nl/)[116] was used to generate all the Venn diagrams of this study.

### Co-immunoprecipitation assays

PDOX-GBM tumors were carefully dissected and placed in cold PBS with protease inhibitors. Tumor or brain tissues were then chopped and transferred to glass tubes containing 37 °C DMEM for homogenization with glass Dounce tissue grinders. Cells were then washed with cold PBS, resuspended, and lysed in RIPA buffer to isolate supernatant of nuclear extracts, which were then pre-cleared with Dynabeads Protein G. Immunoprecipitation was performed by adding guinea pig anti-ASCL1 antibody for 2 h followed by addition of Dynabeads Protein G rotated at 4 °C overnight. Following incubation at 95 °C for 5 min, both immunoprecipitation products and input proteins were extracted and subjected to Western blot electrophoresis (LI-COR Quick Western Kit) for detection using guinea pig anti-ASCL1 and rabbit anti-OLIG2 on nitrocellulose followed by the appropriate Alexa680/IR800 secondary antibodies. A similar co-IP for *Ascl1*-OE and *Ascl1*-CKO tumors, which serve as negative control, was performed using rabbit anti-ASCL1 followed by Western blot detection using rabbit anti-ASCL1 and mouse anti-OLIG2. All Western blots were imaged on an Odyssey scanner (LI-COR Biosciences). Presentation of full scan Western blots are available in the Source Data file. Sources of antibodies are described in Supplementary Table 2.

### Inducing brain tumors in transgenic mice by electroporation of Cre and CRISPR-Cas 9 + gRNA plasmids

Transgenic mouse lines were crossed to carry a combination of floxed and reporter alleles to generate control, *Ascl1*-conditional knock-out (CKO), *Olig2*-CKO, *Ascl1; Olig2*-double CKO; or *Ascl1*-overexpression (OE) tumors. Brain tumors were induced in mouse pups on the day of birth (P0) by injection of approximately 1 μL volume of the following plasmid mix and concentrations (*FUGW-Cre* [2 μg/μL] + *pX330-Cas9+gNf1* [1 μg/μL] + *pX330-Cas9+gPten* [1 μg/μL] + *pX330-Cas9+gTp53* [1 μg/μL]) into the lateral ventricle, followed by electroporation into NPCs/GPs lining the right dorsal SVZ (Supplementary Fig. 1). The guide RNAs used were designed to target the following sequences of each of the three tumor suppressor genes as previously described[34]: *Tp53* 5′-ACAGCCATCACCTCACT GCA-3′, *Pten* 5′-AAAGACTTGAAGGTGTATAC-3′, and *Nf1* 5′-AGTCA GCACCGAGCACAACA-3′. Electroporation was performed using a NEPA21 Super Electroporator and CUY650-P5 electrokinetic tweezers with 5 mm platinum disk electrodes placed diagonally directly above the right lateral ventricle (+disk) and below the jaw (−disk) of mouse pups (Fig. 2a). Each pup underwent electroporation twice with five pulses (100 V, 50 ms duration, 950 ms interval) about 5 minutes apart to ensure efficient force transfer of plasmids into cells in the SVZ and 100% tumor penetrance. A 2:1 concentration ratio of *Cre* to *gRNA* plasmids was used to ensure that tumors were labeled by tdTOM or GFP in the dorsal cortex.

### Brain tumor harvest and survival curves

Brain tumors were harvested from mice at P30, P60, or when severe neurological symptoms (hunching, seizures, etc.) were observed, which was considered the maximal tumor burden based on IACUC review board and was not exceeded. At this point, tumor mice were humanely euthanized as endpoints for survival curves and terminal stage analyses. Neurological symptoms were verified during brain extraction to ensure it is due to the presence of a tumor mass labeled by fluorescent reporter in the right hemisphere rather than other non-tumor-related complications (i.e., hydrocephalus), which can occur in some mice. Longitudinal analyses and survival curves include both male and female tumor-bearing mice from at least three litters per genotype. Statistical significance and median survival were determined by simple survival analysis (Kaplan-Meier, Prism 10) between control versus each experimental tumor group or between males versus females of the same tumor group.

### H&E staining, immunohistochemistry, and EdU detection of brain tumor sections

Two hours before tumor harvest, 5-ethynyl-2′-deoxyuridine (EdU, 1 μg/μL dissolved in sterile PBS), a thymidine analog was injected (10 μg EdU/g body weight) intraperitoneally to label proliferating tumor cells. To harvest brain tumors, mice were intraperitoneally injected with Avertin (2.5 g 2,2,2 tribromoethanol [Aldrich T4,840-2] + 5 mL 2-methy-2-butanol (amylene hydrate [Aldrich 24,048-6] + 200 mL distilled water) to induce anesthesia. Hearts were then perfused with 4% PFA/1XPBS. Brains were extracted and placed in 4% PFA/1XPBS overnight, washed in 1XPBS followed by submersion in 30% sucrose/1XPBS, and then frozen embedded in O.C.T. compound for cryosectioning.

Hematoxylin and eosin (H&E) staining of mouse brain tumors was performed by the Human Tissue Repository core at UNM. For immunohistochemistry, brains were cryosectioned at 40 μm thickness and then blocked as floating sections for one hour with 2% Goat/Donkey blocking solution with 0.3% Triton X-100, followed by overnight incubation in primary antibodies diluted in the blocking solution. The next day, the floating sections were incubated in the appropriate secondary antibodies conjugated to Alexa fluorophores (488, 568, or 647; ThermoFisher). For EdU staining, brain tumor sections were

incubated for 30 minutes in a detection solution containing 100 mM sodium ascorbate, 4 mM copper sulfate, and 12 μM Cy5 in PBS as previously described[117]. The antibodies and reagents used in this study are described in Supplementary Table 2.

## Quantification of tumor cells positive for EdU, ASCL1, OLIG2, or SOX10

Brain sections were imaged using confocal microscopy (Leica TCS SP8). Images were collected using the 20X oil objective at 2048×2048 resolution (individual image size: 581 μm x 581 μm). Imaging was limited to regions within tumors or on tumor margins with the highest density of EdU and reporter (tdTOM or GFP) co-labeled cells. EdU + / reporter + , ASCL1 + /reporter + , and OLIG2 + /reporter+ staining were quantified with IMARIS software. In brief, 581 μm x 581 μm images with an optical density of 3.12 μm were converted from TIFFs to .ims files, and a colocalization channel was created using DAPI and the tumor reporter to limit quantification to DAPI + /reporter+ tumor cells. Using the Spots application, the number of nuclei with an estimated average diameter of 6.8 μm within the colocalization channel was determined. Next, another channel (EdU, ASCL1, OLIG2, SOX10) was added to the Spots application and assessed again using "Classify Spots." This analysis quantified cells that were positive for the reporter and the channels of interest. A total of 15 images per tumor brain at terminal stages (5 images of distinct tumor areas per coronal section across three coronal sections) were analyzed. Fewer images were analyzed for P30 and dCKO brains because of smaller tumor sizes, but at least five images per brain were analyzed.

## Quantification of cellular immunofluorescent levels of ASCL1 and OLIG2 within tumor cells

Sections of control, *Ascl1*-CKO, *Ascl1*-OE, and *Olig2*-CKO brain tumors were simultaneously stained with the same anti-ASCL1 or anti-OLIG2 antibody concentrations and imaged using the same confocal laser parameters. Cellular immunofluorescence intensity for ASCL1 or OLIG2 was calculated as previously described[21,118]. Briefly, single tumor cells within high magnification images (TIFF) with ASCL1 or OLIG2 signals were encircled using the freeform drawing tool in ImageJ software. The area, integrated density, and mean fluorescence of individual cells, along with mean background fluorescence, were measured for calculation of the total corrected cellular fluorescence (TCCF=integrated density − (area of cell x mean background fluorescence)). We measured TCCF for 20 cells per tumor section and three sections per tumor (60 cells total).

## Quantification of tumor cell migration distance across the midline on contralateral corpus callosum

Tile scans of whole coronal brain sections with visible corpus callosum and labeled tumor cells were collected for each brain at three different rostral-caudal levels through the bulk of the tumor. Using ImageJ software, we captured the total length (TL) of the corpus callosum on the contralateral hemisphere starting at the midline, followed by the migration distance (MD) of tumor (reporter+) cells on the contralateral corpus callosum. The distance of tumor migration on the contralateral corpus callosum was then determined by dividing MD/TL and reported as percentage (%) distance migration on the contralateral corpus callosum for each tumor sample. This normalization was used because large tumor masses, such as those in control and *Ascl1*-CKO tumor mice, can drastically alter the size, absolute length, and morphology of the contralateral corpus callosum.

## Single-cell suspension, fluorescence-activated cell sorting (FACS), and library preparation of tumor cells for 10X Genomics Chromium scRNA-seq

Brains of control (tdTOM + , $n = 3$) or *Ascl1*-OE (GFP + , $n = 3$) tumor-bearing mice with neurological symptoms were freshly harvested

and tumors were dissected out in a bath of Hank's Balanced Salt Solution (without calcium or magnesium). Working on ice, tumor tissue was weighed and then transferred to a gentleMACS C tube and dissociated using the Miltenyi Neural Tissue Dissociation Kit (P) and an Octo-Dissociator with Heaters. Following this, cells underwent FACS in the UNM Comprehensive Cancer Center Shared Resource Flow Cytometry facility (for gating of reporter positive sorted tumor cells, see the Supplementary Fig. 6). Tumor cell concentrations and viability were determined using a Cell Countess II FL device (ThermoFisher) before calculating cell counts and loading the suspension into the Next GEM Chip G and Chromium Controller (10x Genomics) per the manufacturer's protocol for genome-scale metabolic models (GEM). The 10x Chromium Next GEM 3′ protocol was used to create 3′ libraries for sequencing at the University of Colorado Anschutz Medical Campus's Genomics Shared Resource Cancer Center using Illumina NovaSEQ 6000 instruments on S4 flow cells. Briefly, cells were lysed and barcoded within each GEM before first strand cDNA synthesis, and also within each GEM. Cells were pooled prior to library completion as described in the manufacturer's protocol. Library quality was assessed after cDNA synthesis and after completion on the BioAnalyzer using a DNA High Sensitivity Chip (Agilent). Before sequencing, Agilent Tape Station 4200 and Invitrogen Qubit 4.0 reagents were used to determine final library concentrations before dilution, normalization, and pooling at 4 nM. qPCR was used to determine cluster efficiency before loading libraries into NovaSEQ devices.

## Single-cell RNA sequencing analyses of tumor cells

Data were demultiplexed and fastq files were generated using bcl2fastq (v2.20.0.422, Illumina) with parameter "--barcode-mismatches" set to 1. Fastq files were aligned and genes/cells were counted against the mouse reference genome (mm10) using CellRanger (v6.0.0, 10xGenomics)[119]. EGFP and tdTomato sequences were appended to the mm10 genome according to the manufacturer's instructions. Approximately 18,000 tdTOM+ control tumor cells and 25,000 GFP+ *Ascl1*-OE tumor cells met quality criteria for further analyses. CellRanger-generated filtered files for these cells were used for downstream analyses.

Seurat (v5.1.0), an R package (v4.3), was used for downstream unsupervised clustering analyses (https://www.R-project.org/)[119,120]. We used scatter (v1.26.1) to identify and remove cells that were outliers for counts, features, and mitochondrial counts[121]. scDblFinder (v1.12.0) was used to identify and remove doublet cells[122]. Data were transformed using SCTransform (v0.3.5)[123] as implemented in Seurat. Data from all samples were integrated using Seurat's standard integration workflow. Principal component analyses and an elbow plot were used to visualize variances and select principal components (1:30). Clusters were determined using the FindNeighbors and FindClusters function with default parameters and the resolution set to 0.4. GBM subtype and cell type annotations were added using Seurat's MetaFeature function. The list of GBM subtype markers was previously described[13]. Cell type assignment was based on a curated list of cell type-specific signature genes (Supplementary Data 7) adopted from previously published datasets[44–50]. Additionally, each of the 40-50 signature genes for astrocyte, oligodendrocyte (OPC, NFOL, MOL), and microglia lineages were individually validated using Brain RNA-seq (brainrnaseq.org) to ensure high expression and specificity to these lineages in the mouse brain. Cell cycle phases were determined using the CellCycleScoring function within Seurat with a list of mouse cell cycle genes from https://hbctraining.github.io/scRNA-seq/lessons/cell_cycle_scoring.html. Differential gene expression between control and *Ascl1*-OE tumors for each cell type was determined using FindMarkers in conjunction with the R package MAST (v1.28.0)[124] as implemented in Seurat. Significant genes were defined by average $\log_2$ fold-change >±0.33 and an adjusted p-value < 0.05. Genes that were expressed in <20% of cells of each cell type

in both tumor groups were excluded. Unionized RNA-seq results for transcripts of each cell type were calculated by AverageExpression within Seurat with the grouping for each cell type for each tumor. Meta module scoring based on previously published GBM cell states was completed using the R package scalop (v1.1.0)[16]. All heatmaps of the Cell Type Unionized RNA-seq were generated using the R package ComplexHeatmap (v2.18.0)[125,126].

## Statistics & Reproducibility
Statistical tests were analyzed using Prism 8 or 9 software (GraphPad). The presence of outliers was assessed with Grubbs' test using Alpha = 0.05. The experiments were not randomized as this study utilized different tumor genotypes for each experimental group. The group size was determined using a power analysis to provide 80% power to detect a 20% change in survival at 5% significance level. The investigators were not blinded to allocation during experiments and outcome assessments since each brain tumor genotype exhibited unique morphological phenotype and reporter expression.

## Reporting summary
Further information on research design is available in the Nature Portfolio Reporting Summary linked to this article.

## Data availability
The scRNA-seq and OLIG2 ChIP-seq data were generated for this study. The scRNA-seq and ASCL1 and OLIG2 ChIP-seq data that support the findings of this study have been deposited in the Gene Expression Omnibus (GEO) under the primary accession numbers GSE247650, GSE152401, and GSE247977, respectively. Source data are provided with this paper.

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

## Acknowledgements

This research was supported by NINDS K22NS09267 & R01NS121660 (T.Y.V.), NIAAA T32 AA 014127 Alcohol Research Training in Neuroscience grant (B.L.M.), Cancer Prevention and Research Institute of Texas RP130464 (J.E.J), and partially supported by the UNM Comprehensive Cancer Center Support Grant NCI P30CA118100, ACS-IRG-21-146-25-IRG, Pilot Projects #1451,1513, UNM Center for Brain Recovery and Repair NIGMS P20GM109089, and the University of Colorado Anschutz Medical Campus Genomics Shared Resource Cancer Center Support Grant P30CA046934. The following UNM Comprehensive Cancer Center Shared Resource facilities, which receive additional support from the State of New Mexico, were essential toward the completion of this research: Analytical/Translational Genomics, Fluorescence Microscopy, Flow Cytometry, Bioinformatics, and Human Tissue Repository. *FUGW-Cre* plasmids were a gift from Dr. Jason Weick's lab at UNM. We thank Karen Klein of Clarus Editorial Services for professional editing and Drs. Nora Perrone-Bizzozero and Fernando Valenzuela for critical reading of the manuscript.

## Author contributions

B.L.M. and T.Y.V. designed the study, and B.L.M. carried out all mouse brain tumor experiments. K.J.B. and B.L.M. performed bioinformatic analyses of scRNA-seq data. L.E.P.B., M.S.K., J.N., R.H.A., Y.L., and C.M.M. harvested brain tumors and performed immunohistochemistry. H.S., E.V., and B.L.M. performed Co-IP experiments, and H.S. characterized plasmid DNA. T.Y.V .and M.D.B. performed ChIP-seq and RKK performed bioinformatic analyses of ChIP-seq and TCGA GBM RNA-seq under supervision of J.E.J., who also contributed to early stage of this study. Q.R.L. provided Olig2-floxed mice and RMB provided PDOX-GBM mice. B.L.M. and T.Y.V. analyzed all data, prepared figures, and wrote the manuscript. All co-authors provided edits and approved the final manuscript.

## Competing interests

The authors declare no competing interests.
