## [Peer Review file · Nature Communications]

Transcription factors ASCL1 and OLIG2 drive glioblastoma initiation and co-regulate tumor cell types and migration

Corresponding Author: Dr Tou Yia Vue

Version 0:

Reviewer comments:

Reviewer #1

(Remarks to the Author)

In this study, the authors demonstrate that OLIG2 and ASCL1, two bHLH transcription factors, act in a combinatorial fashion during GBM initiation in a mouse model. Using ChIP-seq, the authors first show that while Ascl1 binds many few target sites than Olig2 in the genome, the vast majority of Ascl1 target genes are also targets of Olig2. Of these co-bound target sites, 841 genes were de-regulated in TCGA GBM samples and thus likely directly regulated by ASCL1 and OLIG2. Using a GBM mouse model, the authors then assess tumor formation in Ascl1/Olig2 single and double cKOs. Strikingly, GBM formation was dramatically blocked by the double deletion of Ascl1 and Olig2. Finally, the authors characterize the Ascl1-cKO and Olig2-cKO and Ascl1-OE tumors by marker analysis and Ascl1-OE tumors by scRNA-seq. They found based on these joint analyses that Ascl1 OE turns on Olig2 expression, promote proliferation at P30 and not terminal stages and initiate many astrocyte genes. tumors They further show that ASCL1 and OLIG2 share several transcriptional targets, and identify metabolic and protein translation genes as ASCL1 target genes.

Comments

1. Lines 94-98 “De novo motif analyses showed significant enrichment of E-boxes with a “GC” core (CAGCGT) within ASCL1 and OLIG2 shared binding peaks, whereas E-boxes with a variable “GC/TA” core were observed within peaks for OLIG2 only. This difference in E-box binding preference and enrichment of different co-factor DNA motifs may explain the more promiscuous binding of OLIG2 compared to ASCL1”

The canonical E-box is CANNTG – the flanking residues are normally a concatemer. Is CAGCGT supposed to be CAGCTG? Otherwise, it is not a canonical E-box. Figure 1C has the correct e-box sequence in the schematic, so I presume this is a typo.

The statement about the variable “GC/TA” core is not clear. Do the authors mean that the central GC is preferentially TA in OLIG2 only peaks?

Fig. 1C is not fully described and the statement “enrichment of different co-factor DNA motifs may explain the more promiscuous binding of OLIG2 compared to ASCL1” is not clear. I am not sure what the percentages refer to in Fig. 1c? Do the numbers represent the percentage of those motifs in the genome that are occupied by ASCL1 and OLIG2 or OLIG2 alone? What does the more promiscuous binding refer to? Both the combined TFs and OLIG2 alone show preference to 3 sites? Is it just that the sites that OLIG2 binds alone are more abundant in the genome? Clarification of these points should be provided.

2. lines 114-115 “ChIP-seq analyses revealed the presence of ASCL1 and OLIG2 binding peaks at their respective ASCL1 and OLIG1/2 loci”

It does not look like ASCL1 is binding to the ASCL1 locus – i.e., ASCL1 does not auto-regulate its own transcription, which is consistent with developmental findings (Fig. 1F). So the statement above is not accurate as it implies that there is binding at respective loci? Conversely, both ASCL1 and OLIG2 appear to bind the OLIG2 regulatory regions, and other loci. Clarification should be provided.

Please note, from other datasets the assumption is that Ascl1 does not autoregulate itself. There is no “real” binding of Ascl1

in the vicinity of the regulatory regions for its locus. However, in many ASCL1 overexpression data-sets, strong Ascl1 ChIP-seq peaks within the gene are often seen (e.g., publicly available from Philpott's group human GBM data) that may be contamination from the plasmid DNA, given that it covers exactly the coding region.

3. lines 119-121 "Co-immunoprecipitation assay demonstrated that the shared binding between ASCL1 and OLIG2 may be due to a direct protein-protein interaction since OLIG2 was successfully pulled down with an antibody specific for ASCL1 from PDX-GBM cells"

This sentence should refer to Figure 1G,H.

Also, for this experiment and the experiment in Figure 1I, a control IP with an IgG control antibody should be performed to show the specificity of the pulldown (i.e., not a sticky protein).

4. lines 132-135 "Specifically, plasmids expressing Cre-recombinase and Cas 9 + gRNAs37 targeting Nf1, Pten, and Tp53 were injected and then electroporated into NPCs in the dorsal subventricular zone of the right lateral ventricle of Cre-dependent tdTomato (tdTOM) reporter mice (R26RT/T) at birth (P0)"

Evidence that the gRNAs successfully targeted Nf1, Pten, and Tp53 should be provided. Reference 37 uses gRNA targeting Ptch1, and validates three gRNAs with a SURVEYOR assay. Similar proof of concept should be provided herein (or a reference to their validation provided).

5. lines 138-141 "The presence of tdTOM showed that tumors were aggressive and invasive, capable of migrating across the corpus callosum to the contralateral hemisphere, co-expressed high levels of both ASCL1 and OLIG2, and exhibited similar histopathological characteristics to that of GBMs."

Which histopathological characteristics are exhibited?

From which region is the high magnification images in E taken from? Both Ascl1 and Olig2 appear more widespread than in Figure Suppl 1E,F. It would be better to show in the main figure (or in the supplement) the lower magnification images from which Figure 1 panels E,I,M,Q,T were taken.

6. lines 173-175 "Taken together, our findings imply that ASCL1 and OLIG2 function redundantly downstream of driver mutations to transform affected NPCs into proliferating tumor cells, but these transcription factors regulate opposing aspects of tumor cell migration."

Is the model that in the absence of Olig2, Ascl1 can transactivate pro-migratory genes that might not normally be bound? Have the authors looked in their ChIP-seq data for Ascl1 and Olig2 occupancy of Rnd3, which is a known effector of Ascl1 to regulate neuronal migration during development? Analysis of Rnd3 would be informative whether it is bound or not to rule in or rule out this mode of migratory control.

7. From Figure 3, lines 194-196, the authors state: "These findings highlight the general observation that the cellular levels of ASCL1 and OLIG2 are highly dynamic but inversely proportional to each other within tumor cells."

And then

Lines 261-264 "Taken together, these findings revealed that ASCL1 and OLIG2 are differentially dysregulated by the loss of Nf1, Pten, and Tp53 at early stages (P30), but these two transcription factors are able to reciprocally and positively regulate each other's expression to redundantly promote tumor formation and progression."

From the data in Figure 3, the conclusion that potentially could be reached would be that Ascl1 represses Olig2, but to reach that conclusion, the comparisons in Figure 3G should be to the control bar for both the Ascl1 cKO and Ascl1 OE. Olig2 does not appear to be required to regulate Ascl1 expression, but the comparisons in Figure 3F should be between control and the Olig2 cKO to be sure.

How does this data differ from the data presented in Suppl Figure 3? The two datasets should be discussed together instead of in different sections of the manuscript.

One conclusion is they promote each other's expression and the other is that there are reciprocal interactions.

8. Figure 4 and Figure S2. Quantifications should be provided for all claims of co-expression rates.

From this data, the authors conclude that Ascl1-CKO tumors are predominantly "oligodendrogliomas" and therefore, Ascl1 promotes an astroglial-like fate. The scRNA-seq from Ascl1-OE in the subsequent figures 5-7 supports these claims.

However, for Figure 4, cell counts should be performed. Is the Ascl1 cKO subtype a proneural subtype? And Ascl1 OE a classic subtype with astrocytic features?

These findings and conclusions are somewhat unexpected since Ascl1 specifies an oligodendrocyte fate in subsets of neural progenitor cells during embryonic CNS development, and since other studies have found Ascl1 expression in ODG tumour cells. More discussion of Ascl1 and its known roles in glial development could be provided in the discussion section of

the manuscript.

9. Figure 5. It would be interesting to map Rnd3 given the association between Ascl1 and Rnd3 transactivation and that these Ascl1 OE tumors are highly migratory.

Reviewer #2

(Remarks to the Author)

Myers et al. used multiple transgenic mouse strains and single-cell RNA-seq to dissect the roles of Ascl1 and Olig2 in gliomagenesis, and found that these transcription factors are redundantly required for tumor initiation, but differentially regulate glioma cell types: Ascl1 promotes the neural stem cell/astrocyte fate while Olig2 promotes the oligodendrocyte fate, which is quite convincing. The authors also argue that Ascl1 promotes tumor cell migration while Olig2 does not. The findings are of general interest to the field. However, I am not convinced about the conclusions on tumor migration. Several conclusions should be backed up by statistics, and there are a number of studies the authors should have discussed to put their finding into context.

1. The key issue is that how the authors interpret the "migration" phenotype. Since the Ascl1 CKO tumors and Olig2 CKO tumors have drastically different cellular composition (oligodendrocyte-like vs astrocyte-like), the different tumor cell distribution in these two models could simply be explained the differential migratory capacity of these two cell types alone. Without cell-type controlled migration assays, it is not convincing to conclude that these two TFs differentially regulate migration.

2. The authors should refrain from using the term "cancer stem cells" since this is clearly not a cancer stem cell model/study.

3. The authors should discuss about their findings in the context of previous studies that show the fate-switch during gliomagenesis, such as 10.1038/nn.3790, 10.1038/s41422-020-00451-z, and the results from Richard Lu labs. If possible, they could compare/reanalyze their single-cell data set with previously published models.

4. They should provide more detailed analysis to support their cell type assignment in their single-cell analysis.

5. Since Olig2 KO phenotype in glioma models has already been published by Richard Lu group, the authors should consider focusing on the DKO or Ascl1 OE phenotypes, and provide a summary cartoon to highlight the new results.

Reviewer #3

(Remarks to the Author)

The manuscript by Myers et al builds on previous work by the authors focussed on the role of Ascl1 in glioma. In the current study the authors examine the crosstalk between Ascl1 and Olig2 by combining ChiP-seq of 2 PDX models with autoctonous mouse models. They report that Ascl1 and Olig2 co-regulate several genes involved in DNA binding and cell-cycle and can that the two proteins can physically interact. They also characterise mouse tumours in which Ascl1/Olig2 or both are inactivated or Ascl1 overexpressed, reporting opposing phenotypes in single know-outs suggestive of differences in cell migration downstream of Ascl1. Finally, they show that Ascl1 favours astrocytic fate and to a lesser extent NSC fate, alongside proliferation, using IHC analysis in the different models and scRNA-seq analysis of Control and Ascl1-OE tumours.

The topic is interesting and the mouse models used powerful. However, the paper is underdeveloped and the novelty of its findings somewhat limited. The authors themselves previously showed that Ascl1 drives proliferation. The role of Olig2 in driving oligodendrocyte fate is long established (e.g. Lu 2016, Ligon 2007). Several papers have already shown that similar mouse models recapitulate the heterogeneity of the human disease (Yeo 2022, Pathania 2017, McNicholas 2023, Garcia Diaz 2023 to name a few). The finding that Ascl1 might drive invasion is interesting but there is now follow up to that observation or attempt to functionally test the underpinning mechanisms. A main focus is on the transcriptional network driven by Ascl1/Olig2 yet an important experimental cohort, Olig2 overexpression is missing. Several of the figures are very minimalistic and data-light with overreliance on bioinformatics plots. The story is also quite disjointed overall (for example it isn't clear what the results shown in S3 and S4 add to the findings and they are then not followed up or developed further)and many conclusions are overstated. Finally, no attempt is made at validating the findings in the human disease.

Version 1:

Reviewer comments:

Reviewer #1

(Remarks to the Author)

The authors have adequately addressed all of my concerns.

Reviewer #2

(Remarks to the Author)

The authors have properly addressed all of my concerns.

Reviewer #3

(Remarks to the Author)

The revised manuscript by Myers et al, provides very limited new data and minor text changes to address my criticisms. Unfortunately, these revisions fall short of addressing my initial concerns around novelty, limited mechanistic insight and functional validation.

We thank the reviewers for their insightful comments and suggestions on our manuscript. We have carefully edited the manuscript to address all comments and concerns through clarifications within the text as well as included additional data where appropriate. *Major revision or addition to the manuscript is indicated with blue text.* Additionally, we have re-analyzed our single cell RNA-seq data which substantially strengthened the findings of our manuscript. Specifically, based on elbow plot ranking of the variance contributed by each principal component, we determined that true signal was captured within 30 principal components compared to the 40 that were used in our first analysis. This has changed the number of UMAP cell clusters and visualization of our data. Moreover, we removed the neuronal, endothelial and pericyte cell types from our analysis because the cells that were assigned to these identities only expressed about 10% of the 40-50 signature genes for these cell types, suggesting that these are not actual cell lineages within the tumor composition. Lastly, we refined our analysis of differentially expressed gene (DEGs) of our scRNA-seq data by comparing cell type x cell type for the seven cell types (qNSC, aNSC, astrocyte, OPC, NFOL, MOL, micglroia) between control and *Ascl1*-overexpression tumor cells.

Below is our point-by-point response to comments of each reviewer.

Reviewer #1 Primary Concerns:

1. *Lines 94-98 “De novo motif analyses showed significant enrichment of E-boxes with a “GC” core (CAGCGT) within ASCL1 and OLIG2 shared binding peaks, whereas E-boxes with a variable “GC/TA” core were observed within peaks for OLIG2 only. This difference in E-box binding preference and enrichment of different co-factor DNA motifs may explain the more promiscuous binding of OLIG2 compared to ASCL1”.*

The canonical E-box is CANNTG – the flanking residues are normally a concatemer. Is CAGCGT supposed to be CAGCTG? Otherwise, it is not a canonical E-box. Figure 1C has the correct e-box sequence in the schematic, so I presume this is a typo.

This was indeed a typo and is now corrected within the manuscript (Line 99).

The statement about the variable “GC/TA” core is not clear. Do the authors mean that the central GC is preferentially TA in OLIG2 only peaks?

We have clarified in the text that OLIG2 demonstrates an ability to bind more indiscriminately to E-box motifs with some preference for “GC” or “TA” core nucleotides, whereas ASCL1 & OLIG2 shared binding sites demonstrate a stronger preference in binding to E-boxes with a “GC” core (Lines 97-100).

Fig. 1C is not fully described and the statement “enrichment of different co-factor DNA motifs may explain the more promiscuous binding of OLIG2 compared to ASCL1” is not clear. I am not sure what the percentages refer to in Fig. 1c? Do the numbers represent the percentage of those motifs in the genome that are occupied by ASCL1 and OLIG2 or OLIG2 alone? What does the more promiscuous binding refer to? Both the combined TFs and OLIG2 alone show preference to 3 sites? Is it just that the sites that OLIG2 binds alone are more abundant in the genome? Clarification of these points should be provided.

The percentage for **Figure 1C** describes the frequency/probability of finding the indicated DNA motif (i.e. CAGCTG) within ASCL1 & OLIG2 shared binding peaks or OLIG2 only peaks, compared to the (percentage) frequency of that motif in background genomic sequence of similar distance. This clarification is now included in the legend of **Figure 1**.

We have clarified that the co-factor motifs enriched in ASCL1 & OLIG2 shared peaks are different compared to those enriched in OLIG2 only peaks. We believe this difference, combined with ASCL1's strong preference for binding to CAGCTG E-box compared to OLIG2's more indiscriminate binding to CANNTG E-box, are partly the reason why OLIG2 binds to more sites in the genome than ASCL1; otherwise ASCL1 would show a greater overlap with OLIG2 binding sites, particularly those sites where E-boxes are present. To avoid confusion, we have removed "promiscuous binding" and replaced it with "differential binding specificity" to describe the difference in binding between ASCL1 and OLIG2 in the genome of GBMs (Line 97-104).

2. *Lines 114-115 "ChIP-seq analyses revealed the presence of ASCL1 and OLIG2 binding peaks at their respective ASCL1 and OLIG1/2 loci"*

It does not look like ASCL1 is binding to the ASCL1 locus – i.e., ASCL1 does not auto-regulate its own transcription, which is consistent with developmental findings (Fig. 1F). So the statement above is not accurate as it implies that there is binding at respective loci? Conversely, both ASCL1 and OLIG2 appear to bind the OLIG2 regulatory regions, and other loci. Clarification should be provided.

Please note, from other datasets the assumption is that Ascl1 does not autoregulate itself. There is no "real" binding of Ascl1 in the vicinity of the regulatory regions for its locus. However, in many ASCL1 overexpression data-sets, strong Ascl1 ChIP-seq peaks within the gene are often seen (e.g., publicly available from Philpott's group human GBM data) that may be contamination from the plasmid DNA, given that it covers exactly the coding region.

Although there does not appear to be a strong ASCL1 binding peak at the ASCL1 locus, it is nonetheless called as a target of ASCL1 based on GREAT analysis (**Supplementary Table 2 (TS2)**). We have revised the text to make it clear that there is only strong OLIG2 binding peak at the *Ascl1* locus while there are strong shared ASCL1 and OLIG2 binding peaks around the OLIG1/2 loci (Lines 126-129).

3. *Lines 119-121 "Co-immunoprecipitation assay demonstrated that the shared binding between ASCL1 and OLIG2 may be due to a direct protein-protein interaction since OLIG2 was successfully pulled down with an antibody specific for ASCL1 from PDX-GBM cells"*

This sentence should refer to Figure 1G,H.

This is now corrected to match Figure 1. (Lines 131-134)

Also, for this experiment and the experiment in Figure 1I, a control IP with an IgG control antibody should be performed to show the specificity of the pulldown (i.e., not a sticky protein).

The Co-IP assays shown for PDX-GBMs (**Figure 1I**: IP with guinea pig anti-ASCL1, made by Jane Johnson Lab) were done several years ago and are no longer available. To better demonstrate the specificity of the protein-protein interactions between ASCL1 and OLIG2, we performed co-IP for *Ascl1*-OE tumor and *Ascl1*-CKO tumor, which serves as negative control, using a rabbit anti-ASCL1/Mash1 antibody (Abcam – ab211327).

New Figure 3J: We showed that, very similar to the PDX-GBMs, OLIG2 was successfully pulled-down from *Ascl1*-OE tumor protein lysates following ASCL1-IP. However, this was not observed

following ASCL1-IP of protein lysates from *Ascl1*-CKO tumor, although Western immunoblotting clearly showed the presence of OLIG2 in *Ascl1*-CKO tumor as seen in *Ascl1*-OE tumor. This finding demonstrates the specificity of the ASCL1 antibody and supports our interpretation of direct dimerization between ASCL1 and OLIG2 in vivo.

4. *Lines 132-135 “Specifically, plasmids expressing Cre-recombinase and Cas 9 + gRNAs37 targeting Nf1, Pten, and Tp53 were injected and then electroporated into NPCs in the dorsal subventricular zone of the right lateral ventricle of Cre-dependent tdTomato (tdTOM) reporter mice (R26RT/T) at birth (P0).”*

Evidence that the gRNAs successfully targeted Nf1, Pten, and Tp53 should be provided. Reference 37 uses gRNA targeting Ptch1, and validates three gRNAs with a SURVEYOR assay. Similar proof of concept should be provided herein (or a reference to their validation provided).

This specificity of the gRNAs to successfully target *Tp53*, *Nf1*, and *Pten* loci was previously validated in Zuckermann et al., 2013 (the aforementioned reference) through PCR amplification of tumor tissues followed by Sanger sequencing.

New Supplementary Figure 1G: We have characterized tumor incidence and now included a table demonstrating that development of lethal brain tumors with 100% penetrance required electroporation of three gRNAs to delete *Tp53*, *Nf1*, and *Pten*. In contrast, electroporation of only one or any combination of just two gRNAs fails to reliably induce lethal brain tumors even by 6 months of age. This further highlights the specificity of the gRNAs and is now described in the results section (Lines 154-157).

5. *Lines 138-141 “The presence of tdTOM showed that tumors were aggressive and invasive, capable of migrating across the corpus callosum to the contralateral hemisphere, co-expressed high levels of both ASCL1 and OLIG2, and exhibited similar histopathological characteristics to that of GBMs.”*

Which histopathological characteristics are exhibited?

New Supplementary Figure 1H: We have included H&E staining showing the histopathological characteristics of GBM such as hypercellularity, pseudopalisading necrosis, mitotic cells, multinucleated giant cells, and atypical nuclei visible within control tumors at terminal stage. This also describe in the manuscript (Lines 157-160).

From which region is the high magnification images in E taken from? Both Ascl1 and Olig2 appear more widespread than in Figure Suppl 1E,F. It would be better to show in the main figure (or in the supplement) the lower magnification images from which Figure 1 panels E,I,M,Q,T were taken.

The region where the high magnification images of ASCL1 and/or OLIG2 in **Figure 2E,I,M,Q,T** and **Supplementary Figure 1E,F** were taken from are now indicated with asterisks in their corresponding low magnification images of **Figure 2D,H,L,P,S**, and **Supplementary Figure 1C**.

ASCL1 and OLIG2 is less prevalent in **Supplementary Figure 1E,F** because these images are from P6 brains, which were only electroporated with Cre and did not include the CRISPR gRNAs to delete the tumor suppressor genes. This **Supplementary Figure 1** is meant to demonstrate that the tdTOM+ electroporated cell-of-origin are likely to be radial glia, as evidence by the

presence of radial processes extending into the cortical plate and the lack of expression of ASCL1 and OLIG2, which are markers of glial progenitor and precursor cells. This point is now clarified in the manuscript (Lines 149-151).

6. *Lines 173-175 “Taken together, our findings imply that ASCL1 and OLIG2 function redundantly downstream of driver mutations to transform affected NPCs into proliferating tumor cells, but these transcription factors regulate opposing aspects of tumor cell migration.”*

Is the model that in the absence of Olig2, Ascl1 can transactivate pro-migratory genes that might not normally be bound?

Based on our finding that *Olig2*-CKO and *Ascl1*-OE tumors are more migratory than control and especially *Ascl1*-CKO tumors, it is our model that in the absence of OLIG2 or imbalance of high levels of ASCL1 to lower levels of OLIG2, ASCL1 becomes more readily available to turn on pro-migratory genes. We proposed in the discussion that OLIG2 may repress ASCL1's pro-migratory function in part through direct dimerization with ASCL1. Thus, in tumors in which OLIG2 dominates (control & *Ascl1*-CKO), ASCL1 is recruited by OLIG2 away from pro-migratory genes to bind to oligodendrocyte lineage genes (Lines 517-528).

New Supplementary Figure 6: We now provide direct evidence of ASCL1's pro-migratory function in P4 brains in which induction of *Ascl1*-OE prematurely induces migration of newly transformed tumor cells out of the subventricular zone into the striatum, cortex, and corpus callosum compared to control tumor cells, which mostly do not express ASCL1 at this stage. This is described in the results section (Lines 296-306).

Have the authors looked in their ChIP-seq data for Ascl1 and Olig2 occupancy of Rnd3, which is a known effector of Ascl1 to regulate neuronal migration during development? Analysis of Rnd3 would be informative whether it is bound or not to rule in or rule out this mode of migratory control.

In our ChIP-seq data, we found that *RND3* is a target of both ASCL1 and OLIG2; however, the expression of *RND3* was not positively correlated with expression of either *ASCL1* or *OLIG2* in human GBMs (**TS3, TS4**). Additionally, we do not see significant upregulation of *Rnd3* in our *Ascl1*-OE tumors (**TS8**), which suggests that ASCL1 likely uses different downstream mechanisms to promote tumor cell migration than what is seen developmentally in neurons. We have included this information in the discussion (Lines 540-546).

7. *From Figure 3, lines 194-196, the authors state: “These findings highlight the general observation that the cellular levels of ASCL1 and OLIG2 are highly dynamic but inversely proportional to each other within tumor cells.”*

And then

Lines 261-264 “Taken together, these findings revealed that ASCL1 and OLIG2 are differentially dysregulated by the loss of Nf1, Pten, and Tp53 at early stages (P30), but these two transcription factors are able to reciprocally and positively regulate each other's expression to redundantly promote tumor formation and progression.”

From the data in Figure 3, the conclusion that potentially could be reached would be that Ascl1 represses Olig2, but to reach that conclusion, the comparisons in Figure 3G should be to the control bar for both the Ascl1 cKO and Ascl1 OE. Olig2 does not appear to be required to

regulate Ascl1 expression, but the comparisons in Figure 3F should be between control and the Olig2 cKO to be sure.

How does this data differ from the data presented in Suppl Figure 3? The two datasets should be discussed together instead of in different sections of the manuscript.

We have performed the proper comparison of immunofluorescent cellular levels of ASCL1 or OLIG2 between control versus their respective experimental tumors for **Figure 3F,G**, and results of statistical analyses are now included in the figure and manuscript (Lines 219-224).

We have also clarified in the text that **Figure 3F,G** measures the cellular levels of ASCL1 and OLIG2 within the tumor cells whereas **Supplementary Figure 3** (now **Supplementary Figure 2**) measures the proportion of tumor cells that are ASCL1+, OLIG2+, or double ASCL1+;OLIG2+ . We have rearranged these figures so that these data are now presented together (Lines 219-235).

8. *Figure 4 and Figure S2. Quantifications should be provided for all claims of co-expression rates.*

New Supplementary Figure 2M: We have included a quantification of the percentage of ASCL1+ cells that are OLIG2+ and vice versa.

From this data, the authors conclude that Ascl1-CKO tumors are predominantly “oligodendrogliomas” and therefore, Ascl1 promotes an astroglial-like fate. The scRNA-seq from Ascl1-OE in the subsequent figures 5-7 supports these claims.

However, for Figure 4, cell counts should be performed. Is the Ascl1 cKO subtype a proneural subtype? And Ascl1 OE a classic subtype with astrocytic features?

These findings and conclusions are somewhat unexpected since Ascl1 specifies an oligodendrocyte fate in subsets of neural progenitor cells during embryonic CNS development, and since other studies have found Ascl1 expression in ODG tumour cells. More discussion of Ascl1 and its known roles in glial development could be provided in the discussion section of the manuscript.

New Supplementary Figure 4: We have included quantification of the percentage of tumor cells that are SOX10+ for control, *Ascl1*-CKO, *Olig2*-CKO, *Ascl1;Olig2*-dCKO, and *Ascl1*-OE tumors. We also included whole brain section images of staining for SOX10 and GFAP for all these tumor types. These data clearly showed the astrocytoma (*Olig2*-CKO, dCKO), oligodendroglioma (*Ascl1*-CKO), and mixed glioma (control, *Ascl1*-OE) phenotypes of these tumors.

We have not determined the subtype of the *Ascl1*-CKO tumors using gene expression. Based on SOX10 expression and the oligodendrocyte lineage fate of *Ascl1*-CKO tumor cells, these tumors appear to be proneural. Interestingly, based on our scRNA-seq data, we have determined that overexpression of *Ascl1* promotes a more proneural subtype at the expense of the classical subtype, despite the reduced expression of oligodendrocyte lineage genes and SOX10+ tumor cells. However, this may be due to *Ascl1*, as well as targets of ASCL1, being listed as markers of the proneural subtype. We have clarified this within the discussion (Lines 549-560).

Additionally, we have included a schematic summary in the discussion describing known roles of ASCL1 and OLIG2 in gliogenesis in the cortex contrast with gliomagenesis of the various tumor types (Lines 431-461).

9. *Figure 5. It would be interesting to map Rnd3 given the association between Ascl1 and Rnd3 transactivation and that these Ascl1 OE tumors are highly migratory.*

We have addressed this along with item #6.

Reviewer #2 Primary Concerns:

1. *The key issue is that how the authors interpret the "migration" phenotype. Since the Ascl1 CKO tumors and Olig2 CKO tumors have drastically different cellular composition (oligodendrocyte-like vs astrocyte-like), the different tumor cell distribution in these two models could simply be explained the differential migratory capacity of these two cell types alone. Without cell-type controled migration assays, it is not convincing to conclude that these two TFs differentially regulate migration.*

We agree that the "migration" phenotype of oligodendrocyte-like vs astrocyte-like tumors could be partly due to the natural migratory capacity of the cell type of these tumors. However, the fact that astrocyte-like *Olig2*-CKO tumor cells, whether at P30, P60, or terminal stages, exhibit more migratory capacity than *Ascl1*;*Olig2*-CKO tumor cells, which are also astrocyte-like, is a strong indication that ASCL1 plays a prominent role in tumor migration. The role of ASCL1 in tumor migration may also hold true in oligodendrocyte-like tumor cells which express ASCL1. Indeed, ASCL1+ control tumors are more migratory than *Ascl1*-CKO tumors (**Figure 3I**), although both these tumor types appear to exhibit similar number of SOX10+ oligodendrocyte-like tumor cells (**Supplementary Figure 4U**). To better demonstrate ASCL1's role in tumor cell migration, we tested the effect of *Ascl1* overexpression on newly transformed cells prior to their cell type specification.

New Supplementary Figure 6: We showed that 4 days after tumor induction (P4), *Ascl1*-OE was sufficient to promote extensive migration of newly transformed GFP+ tumor cells out of the subventricular zone into surrounding brain regions compared control tdTOM+ tumor cells at P4. At this stage, both GFP+ and tdTOM+ tumor cells were similarly positive for GFAP and completely negative for SOX10. The similarity in morphology and co-localization with GFAP suggests that ASCL1 directly promotes migration independent of cell type.

2. *The authors should refrain from using the term "cancer stem cells" since this is clearly not a cancer stem cell model/study.*

We agree with the reviewer and therefore have removed mentions of cancer stem within the results and discussion unless specifically discussing cancer stem cell research.

3. *The authors should discuss about their findings in the context of previous studies that show the fate-switch during gliomagenesis, such as 10.1038/nn.3790, 10.1038/s41422-020-00451-z, and the results from Richard Lu labs.*

We have dedicated a section in the discussion to contrast our findings of fate-switching of glioma cells in our tumor mouse model with these earlier studies (Lines 484-512).

If possible, they could compare/reanalyze their single-cell data set with previously published models.

We agree that there should be more discussion of how our data fits into the recently published analyses of GBMs through scRNA-seq. To this end, we have overlaid our cells onto the GBM cell states previously published by Neftel et al., 2019.

New Supplementary Figure 7: We assigned our tumor cells to the previously published GBM cell states and found that our control tumors exhibit 3X the number of cells that score highly for OPC-like cell states and overexpression of *Ascl1* results in more cells scoring highly for NPC-like and AC-like cellular states. Similarly, we used these gene lists to demonstrate that our unionized RNA-seq analysis shows upregulation of OPC-like genes within the control tumors and upregulation of AC- and NPC-like genes within the *Ascl1*-OE tumors. This is now described in the results section (Lines 380-388).

4. They should provide more detailed analysis to support their cell type assignment in their single-cell analysis.

We have included more information in the results (Lines 335-344) and methods (Lines 780-784) about how we created our cell type signature genes. Briefly, we used scRNA-seq data from quiescent and active NSCs from the adult SVZ of mice as well as scRNA-seq and bulk RNA-seq of astrocytes, oligodendrocytes, and microglia isolated juvenile and adult mouse brains in order to identify cell type specific genes (Codega et al., 2014; Llorens-Bobadilla et al., 2015; Xie et al., 2020; Farhy-Tselnicker et al., 2021; Marques et al., 2016; Marques et al., 2018; Zhang et al., 2014). We also further revised our gene lists to ensure that all genes were cell type-specific and highly expressed within their respective cell types using BrainRNAseq.org, which resulted in some changes to cell type enrichment within our results (Lines 345-361).

5. Since Olig2 KO phenotype in glioma models has already been published by Richard Lu group, the authors should consider focusing on the DKO or Ascl1 OE phenotypes, and provide a summary cartoon to highlight the new results.

We acknowledge in the manuscript that *Olig2*-CKO glioma model has been done by Dr. Richard Lu's group (who is now co-author on the manuscript for providing the *Olig2*-floxed mice). However, they used a very different model of tumor induction and cell-of-origin (from adult white matter OPCs) which resulted in tumors that seem to express lower levels of ASCL1 (based on their bulk RNA-seq). More importantly, according to Dr. Richard Lu, their *Olig2*-CKO tumors show no significant tumor cell migration phenotype as we saw with our *Olig2*-CKO model. Because of these differences, it is critically important that both single and double CKOs and *Ascl1*-OE must be analyzed in our GBM model to better understand the combinatorial function of ASCL1 and OLIG2 in gliomas.

New Figure 9: We have now added a schematic summary figure to further clarify the novel findings of our tumor models in the differing genotypes.

Reviewer #3 Primary Concerns:

1. The topic is interesting and the mouse models used powerful. However, the paper is underdeveloped and the novelty of its findings somewhat limited. The authors themselves previously showed that Ascl1 drives proliferation. The role of Olig2 in driving oligodendrocyte fate is long established (e.g. Lu 2016, Ligon 2007). Several papers have already shown that

similar mouse models recapitulate the heterogeneity of the human disease (Yeo 2022, Pathania 2017, McNicholas 2023, Garcia Diaz 2023 to name a few).

We believe our study provides one of the most complete stories on the combinatorial roles of bHLH transcription factors in regulating glioma tumor initiation, proliferation, migration/invasion, and cell types in vivo. This is extremely important, highly novel, and groundbreaking. Although scRNA-seq has been done for GBMs of other mouse models, the purity and number of our tumor cells and experimental conditions offers invaluable insights on the transcriptome of gliomas that has not previously been uncovered.

*2. The finding that *Ascl1* might drive invasion is interesting but there is now follow up to that observation or attempt to functionally test the underpinning mechanisms.*

We agree that ASCL1's role in driving tumor invasion is very interesting. We provide new data (**New Supplementary Figure 5**) demonstrating that ASCL1 overexpression is sufficient to promote early tumor cell migration. We have initiated several collaborations and plan to follow up on many of the genes that we found are upregulated in *Ascl1*-OE tumors that may directly or indirectly contribute to the highly invasive phenotype of these tumor cells. However, these experiments and studies are well beyond the scope of this already highly dense study.

*3. A main focus is on the transcriptional network driven by *Ascl1/Olig2* yet an important experimental cohort, *Olig2* overexpression is missing. Several of the figures are very minimalistic and data-light with overreliance on bioinformatics plots. The story is also quite disjointed overall (for example it isn't clear what the results shown in S3 and S4 add to the findings and they are then not followed up or developed further) and many conclusions are overstated. Finally, no attempt is made at validating the findings in the human disease.*

We agree with the reviewer that including an *Olig2* overexpression cohort could be interesting. We attempted multiple routes of *Olig2* overexpression by electroporating piggy-bac transposase elements into radial glia in the subventricular zone at P0 with different promoter driving *Olig2*, or just *pCAGG-Olig2* plasmids. Unfortunately, none of these attempts resulted in consistent sustained expression of detectable levels of OLIG2 in electroporated labeled cells that was sufficient to force oligodendrocyte lineage specification. The ideal approach would be to use a mouse line to the *Ascl1*-OE (i.e. *TetO-Olig-ires-GFP*) to be paired with Cre-dependent *Rosa26-LSL-tTA* to reliably drive *Olig2* overexpression. Currently, this mouse line does not exist and cannot be included in this study. Nonetheless, this does not preclude the importance and novel findings of our study.

We have rearranged some of the figures and revise the text to improve the flow of the manuscript, and included additional data to support our conclusions.

To demonstrate that our models and findings have relevance to human disease, we utilized cancer cell-state assignments that were published by Neftel et al 2019 using human GBMs. This is addressed in new **Supplementary Figure 7** as described in point #3 for Reviewer 2 above. We also discussed how our findings directly validate or support previous findings in human GBMs (Lines 560-582).